# Predicting Drug Repurposing Candidates and Their Mechanisms from A Biomedical Knowledge Graph

## Abstract

Computational drug repurposing is a cost- and time-efficient method to identify new indications of approved or experimental drugs/compounds. It is especially critical for emerging and/or orphan diseases due to its cheaper investment and shorter research cycle compared with traditional wet-lab drug discovery approaches. However, the underlying mechanisms of action between repurposed drugs and their target diseases remain largely unknown, which is still an unsolved issue in existing repurposing methods. As such, computational drug repurposing has not been widely adopted in clinical settings. In this work, based on a massive biomedical knowledge graph, we propose a computational drug repurposing framework that not only predicts the treatment probabilities between drugs and diseases but also predicts the path-based, testable mechanisms of action (MOAs) as their biomedical explanations. Specifically, we utilize the GraphSAGE model in an unsupervised manner to integrate each entity's neighborhood information and employ a Random Forest model to predict the treatment probabilities between pairs of drugs and diseases. Moreover, we train an adversarial actor-critic reinforcement learning model to predict the potential MOA for explaining drug purposing. To encourage the model to find biologically reasonable paths, we utilize the curated molecular interactions of drugs and a PubMed-publication-based concept distance to extract potential drug MOA paths from the knowledge graph as "demonstration paths" to guide the model during the process of path-finding. Comprehensive experiments and case studies show that the proposed framework outperforms state-of-the-art baselines in both predictive performance of drug repurposing and explanatory performance of recapitulating human-curated DrugMechDB-based paths.

## 1 Introduction

Traditional drug development is a time-consuming process (from initial chemical identification to clinical trials and finally to FDA approval) that takes around 10-15 years and also comes along with billions-of-dollars investments and high failure rates (Berdigaliyev & Aljofan, 2020). Considering the rapid pace of novel disease evolution, it is urgent to find a more efficient and economical drug discovery method. Fortunately, it has been observed that a single drug can often be effective in treating multiple diseases. For example, thalidomide was originally used as an anti-anxiety medication (Miller, 1991), and was later found to have the potential for the treatment of cancers (Singhal et al., 1999). Hence, drug repurposing, also known as the identification of new uses for the approved or experimental drugs/compounds, might bring us the hope to address this urgent need with the advantage of a shorter research cycle, lower investments, and more pre-existing safety tests.

Existing drug repurposing approaches can roughly be categorized into three groups: experimental-based approaches (e.g., binding affinity assays, phenotypic screening), clinical-based approaches (e.g., off-label drug use analysis), and computational-based approaches (e.g., network-based approaches) (Dhir et al., 2020). Due to the advancement of techniques, more and more publicly available biomedical data can be freely accessed in different databases such as DrugBank (Wishart et al., 2017), ChEMBL (Gaulton et al., 2012), HMDB (Wishart et al., 2018), which makes the computational approaches seem to be more cost-efficient, particularly when the goal is to prioritize repur-

posed targets for followup experimental investigation. One of the computational drug repurposing methods commonly used in recent years is to integrate existing biomedical relations from databases or literature into a so-called **biomedical knowledge graph** (BKG) where unknown drug-disease treatment relationships are predicted via different knowledge graph (KG)-based machine learning models (Himmelstein et al., 2017; Ioannidis et al., 2020b; Zhang et al., 2021; Zhang & Che, 2021). Although these KG-based models are demonstrated to have good predictive performance for drug repurposing, they struggle to explain why some drugs can be useful for treating a given disease in an intuitive and easy-to-understand fashion. To solve the "black-box" concern for drug repurposing prediction, some methods are proposed to leverage KG-based paths as explanations, as illustrated in Figure 1. However, these existing models cannot be efficiently applied to a large and general BKG without additional weighted edge information (Sosa et al., 2020) or pre-defined meta-paths derived from domain experts or inefficient computational methods (e.g., degree-weighted path count – "DWPC") (Liu et al., 2021).

In this study, we customize a large and standardized biomedical knowledge graph and propose a computational drug repurposing framework that predicts not only the treatment probabilities between drugs and diseases but also the KG-based **mechanism of action** (MOA) (Davis, 2020) paths as their biomedical explanations based on the treatment predictions. For drug repurposing predictions, we first calculate attribute embedding as the initial feature of each node and employ the GraphSAGE model in an unsupervised manner to further capture the neighborhood information for each node, then a Random Forest model is utilized to predict the treatment probability of drug-disease pairs based on their embeddings. To predict the MOA paths, we employ the ADversarial Actor-

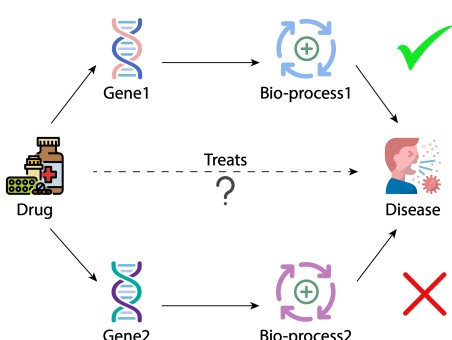

Figure 1: Drug repurposing prediction and path-based explanation.

Critic (ADAC) reinforcement learning (RL) model (Zhao et al., 2020) to perform path-finding on the knowledge graph. To encourage the RL model to find paths that are biologically reasonable, we amplify it with knowledge-and-publication-based "demonstration paths", paths that explain why a drug can treat a disease. Although the underlying mechanisms of action between repurposed drugs and their target diseases largely remain vague, in this study, we assume that a repurposed drug follows similar molecular mechanisms as the known MOAs to treat different diseases. Based on this assumption, we define demonstration paths based on the known drug-target interactions from a curated drug database (e.g., DrugBank v5.1 (Wishart et al., 2017)) and a chemical-knowledge-centric data provider (e.g., Molecular Data Provider v1.2 [1]) as well as an adjusted PubMed-publication-based version of Normalized Google Distance (NGD) (Cilibrasi & Vitanyi, 2007). In summary, the main contributions are summarized as follows:

- We propose a novel computational model framework that both accurately predict how likely a drug can be used to treat a disease and also predict its corresponding knowledge graph-based mechanism of action path as the explanation of the predicted treatment.

- We are innovative in using a knowledge-based and publication-based method to extract demonstration paths from a BKG and leverage it to guide the RL model to identify biologically reasonable paths. Empirical results demonstrate the great effectiveness of it.

By comparing with the existing popular KG-based models and evaluating the predicted paths with an expert-curated path-based drug MOA database *DrugMechDB* (Mayers et al., 2020), we show that this proposed model framework outperforms the state-of-the-art baseline models on both the predictive performance of drug repurposing and the explanatory performance of recapitulating human-curated MOA paths provided by DrugMechDB. In further case studies, by comparing the model predictions with the real regulatory networks, we show that the proposed framework is effective in identifying biologically reasonable KG-based paths for real-world applications.

---

[1] https://github.com/NCATSTranslator/Translator-All/wiki/Molecular-Data-Provider

## 2 RELATED WORK

**Biomedical knowledge graphs**   A biomedical knowledge graph (BKG) is normally defined as a heterogeneous, semantic graph that integrates biomedical information from multiple data sources (Nicholson & Greene, 2020). Many open-source BKGs have been developed for different biomedical research purposes. These existing BKGs can be categorized into three groups: database-based BKGs, literature-based BKGs, and mixed BKGs. The database-based BGKs (e.g., *Hetionet* (Himmelstein et al., 2017), *BioKG* (Walsh et al., 2020), *CBKH* (Su et al., 2021)) are constructed with biomedical data and their relations stored in existing biological databases. The literature-based BKGs are obtained by leveraging Natural Language Processing (NLP) techniques to extract the semantic relations from a large amount of available biomedical literature and electronic health record (EHR) data. They are more disease-specific if the text sources are only focused on specific diseases. For example, Zhang et al. (2021) recently constructed a literature knowledge graph via the collections of Covid-19 literature from the Semantic MEDLINE Database (SemMedDB) (Kilicoglu et al., 2012) as well as other Covid-19 datasets for Covid-19 drug predictions. The mixed BKGs (e.g., *CKG* (Santos et al., 2020), *DRKG* (Ioannidis et al., 2020a)) are generated by combining the knowledge collected from both existing biological databases and available text-based data.

**KG-based computational methods for drug repurposing**   KG-based computational approaches have been widely used in drug repurposing in recent years (Zhang et al., 2021; Al-Saleem et al., 2021; Yan et al., 2021). These approaches treat drug repurposing as a link prediction task by applying different machine learning models to biomedical knowledge graphs. These models can mainly be classified into three groups: tensor-factorization-based models, translation-based models, and neural network-based models (Zeng et al., 2022). The tensor-factorization-based models (Nickel et al., 2011; Yang et al., 2014) consider a KG as a cubic tensor in which three dimensions respectively represent head entity, tail entity, and their relations and then leverage tensor-factorization methods (Rabanser et al., 2017) to recover the missing relations in KGs. The translation-based models (Bordes et al., 2013; Lin et al., 2015; Sun et al., 2019) regard the semantic relations in KGs as a "translation" process, that is, given a triple (head entity, relation, tail entity) the head entity can be translated to the tail entity via the relation-specific "translation". The neural network-based models (Ioannidis et al., 2020b; Dettmers et al., 2017) leverage graph convolutions or 2D convolutions to learn the underlying relationships between two entities.

**Explanation of drug repurposing**   Although drug repurposing prediction can accelerate the process of drug discovery, one of the greatest concerns in computational drug repurposing is the lack of biologically reasonable explanations, which hinders its wide adoption in clinical settings. Currently, there are few computational models designed for drug repurposing explanations. A common and intuitive explanation for drug repurposing leverages the semantic KG-based paths between given drug-disease pairs. To the best of our knowledge, existing models either utilize known statistics scores or use reinforcement learning models to find the most likely KG-based paths for explaining drug repurposing. For example, Sosa et al. (2020) applied a graph embedding model UKGE (Chen et al., 2019), which utilizes the confidence scores of relation edges in a literature-based KG *GNBR*, to identify new indications of drugs for rare diseases and then explain the results via the highest-ranking paths based on confidence scores. Liu et al. (2021) developed a Reinforcement Learning-based model "PoLo" that utilizes the biological meta-paths identified in Himmelstein et al. (2017) via a computationally inefficient method "DWPC" to supervise path searching for drug repurposing.

## 3 KNOWLEDGE GRAPH

To accommodate biomedical-reasonable predictions of drug repurposing and mechanism of action, the ideal knowledge graph should accurately represent comprehensive and diverse interactions among known biological entities. Thus, we utilize the canonicalized version of the Reasoning Tool X Knowledge Graph 2 (*RTX-KG2c*) (Wood et al., 2022), one of the largest open-source biomedical knowledge graph (BKG) that has been widely used in the Biomedical Data Translator Project (Translator Consortium, 2019b;a). Compared to other commonly used open-source BKGs mentioned in

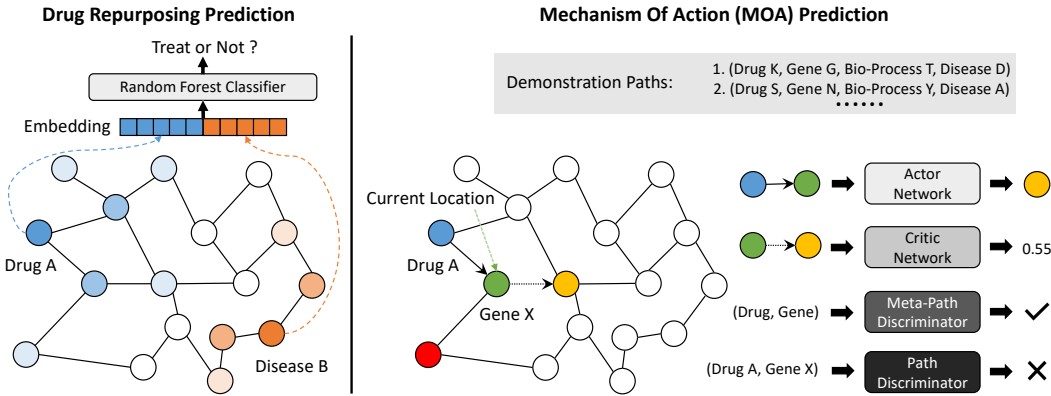

Figure 2: Illustration of the drug repurposing prediction model (left, see Sec. 4.1 for more details) and mechanism of action (MOA) prediction model (right, see Sec. 4.2 for more details).

Sec. 2, *RTX-KG2c* is biolink-model-based [2] standardized (Unni et al., 2022) and regularly-updated. The version 2.7.3 of *RTX-KG2c* that we use aggregates data from 70 public knowledge sources into a large graph where all biological concepts (e.g., "ibuprofen") are represented as vertices and all concept-predicates-concepts (e.g., "ibuprofen - increases activity of - GP1BA gene") are presented as edges. We customized it for drug repurposing purpose with four principles: 1). excluding the nodes whose categories are irrelevant to drug repurposing explanation (e.g., "GeographicLocation" and "Device"); 2). filtering out the low-quality edges based on our criteria; 3). removing the hierarchically redundant edges; 4). excluding all drug-disease edges (see more details in Appx. A.1). After this, 3,659,165 nodes with 33 distinct categories and 18,291,237 edges with 74 distinct types are left in our customized knowledge graph, which is used for downstream model training.

# 4 PROPOSED METHOD

In this section, we describe our proposed model framework (see Figure 2) that predicts drug repurposing and identifies the possible KG-based mechanism of action (MOA) paths as its biological explanations. The proposed framework consists of two components: drug repurposing prediction with GraphSAGE (Hamilton et al., 2017) plus a Random Forest (Sec. 4.1) and MOA prediction with an adversarial actor-critic reinforcement learning (RL) (Sec. 4.2).

**Notation**: Let $\mathcal{G} = \{\mathcal{V}, \mathcal{E}\}$ be a directed knowledge graph, where each node $v \in \mathcal{V}$ represents a particular biomedical entity (e.g., a specific drug, disease, or gene, etc.) and each edge $e \in \mathcal{E}$ represents a biomedical relationship (e.g., *interacts-with*). We use $\mathcal{V}^{\text{drug}}$ to represent all the drug nodes and $\mathcal{V}^{\text{disease}}$ to represent all the disease nodes (defined in item 4 in Appx. A.1).

## 4.1 DRUG REPURPOSING PREDICTION

Drug repurposing aims to identify new indications of approved or experimental drugs. We solve it as a link prediction problem on the knowledge graph $\mathcal{G}$. Specifically, given any drug-disease pair $(v_i, v_j)$ where $v_i \in \mathcal{V}^{\text{drug}}$ and $v_j \in \mathcal{V}^{\text{disease}}$, we predict the probability that drug $i$ can be used to treat disease $j$. We first use GraphSAGE to calculate the embedding for each node. Ideally, the node embeddings should contain two kinds of information: node attributes and node neighborhoods. To capture the neighborhood information, we optimize GraphSAGE to encourage neighbor nodes to have similar embeddings and non-neighbor nodes to have distinct embeddings. Specifically, we perform random walks for each node to collect its neighborhood information and train the model to maximize a node's similarity with its neighbor nodes. For a node $u$, the loss is calculated as:

$$J_{\mathcal{G}}(\boldsymbol{z}_u) = -\log(\sigma(\boldsymbol{z}_u^{\top}\boldsymbol{z}_v)) - k \cdot \mathbb{E}_{v_n \sim P_n(v)} \log(\sigma(-\boldsymbol{z}_u^{\top}\boldsymbol{z}_{v_n})) \tag{1}$$

---

[2]Biolink model (https://biolink.github.io/biolink-model) is a standardized BKG ontology framework

where $\boldsymbol{z}_u, \boldsymbol{z}_v$ are respectively the embeddings of nodes $u, v$, $\sigma$ is the sigmoid function, $v$ is a node that co-occurs with $u$ in fixed-length random walks, $P_n$ represents negative sampling distribution, and $k$ indicates the number of negative samples (nodes not in $u$'s fixed-length neighborhood).

To capture the node attributes information, we utilize the PubMedBERT model (Gu et al., 2022), a pre-trained language model designed for biomedical texts, to generate a node attribute embedding for each node based on the concatenation of the node's name and category. We further compress the embeddings to 100 dimensions with Principal Components Analysis (PCA) to reduce memory usage and use them as the initial node feature for GraphSAGE. In this way, the final GraphSAGE embedding of each node contains the information of both graph topology and node attributes. We concatenate the GraphSAGE embeddings of drug-disease pairs and use them as input into a Random Forest model to classify each drug-disease pair into one of the "not treat", "treat" and "unknown" classes. We obtain "treat" and "not treat" drug-disease pairs from biomedical datasets (see Appx. A.3.1) and generate "unknown" drug-disease pairs through negative sampling (Mikolov et al., 2013).

## 4.2 MECHANISM OF ACTION (MOA) PREDICTION

When potential indications of a given drug are identified by the proposed drug repurposing model, a natural yet important question is: can we biologically explain the predictions? We solve this by employing a reinforcement learning (RL) model to predict the KG-based MOA paths, which are essentially the paths on the knowledge graph from drug nodes to disease nodes. These KG-based MOA paths can semantically describe an abstract biological process of how a drug treats a disease.

### 4.2.1 DEMONSTRATION PATHS

To encourage the RL agent to terminate the path searching at the expected diseases through a biologically reasonable path, we leverage so-called demonstration paths, a set of biologically likely paths (e.g., drug1-gene1-protein3-disease1), that explains the underlying reasons of why a drug can treat a disease. We extract the demonstration paths by using the known drug-target interactions collected from two curated biomedical data sources: DrugBank (v5.1) and Molecular Data Provider (v1.2) along with a literature-based source that utilizes the adjusted Normalized Google Distance (NGD) applied to concepts appearing in PubMed publication abstracts (see Appx. A.2 for more details).

### 4.2.2 ADAC-BASED REINFORCEMENT LEARNING MODEL

We formulate the MOA prediction as a path-finding problem and adapt the ADversarial Actor-Critic (ADAC) Reinforcement Learning model to solve it. The reinforcement learning is defined as a Markov Decision Process (MDP) which contains:

**States**: Each state is defined as $s_t = \big(v_{drug}, v_t, (v_{t-1}, e_t), \ldots, (v_{t-K}, e_{t-(K-1)})\big)$ where $v_{drug} \in \mathcal{V}^{\mathrm{drug}}$ is a given starting drug node; $v_t \in \mathcal{V}$ represents the node where the agent locates at time $t$; the tuple $\big(v_{t-K}, e_{t-(K-1)}\big)$ represents the previous $K$th node and $(K-1)$th predicate. For the initial state $s_0$, the previous nodes and predicates are substituted by a special dummy node and predicate. We concatenate the embedding of all nodes and predicates of $s_t$ to get the state embedding, where the node embeddings are node attribute embeddings generated with the PubMedBERT model (see Sec. 4.1) and the predicate embeddings employ one-hot vectors.

**Actions**: The action space $A_t$ of each node $v_t$ includes a self-loop action and the actions to reach its outgoing neighbors in the graph $\mathcal{G}$. Due to memory limitation and extremely large out-degree of certain nodes in the knowledge graph, we prune the neighbor actions based on the PageRank scores if a node has neighbors more than a certain threshold. Specifically, we let $A_t = \mathrm{vcat}(\vec{a}_{self}, \vec{a}_1, \ldots, \vec{a}_k, \ldots, \vec{a}_{n_{v_t}})$ where vcat indicates the vertical concatenation and $n_{v_t}$ is out-degree of node $v_t$, $v_t \in \mathcal{V}$. For each action $a_t = (v_t, e_t)$, we concatenate its node and predicate embeddings to obtain action embeddings. We learn two embedding matrices $E^{N_{\mathrm{n}} \times \mathrm{d}}$ and $E^{N_{\mathrm{p}} \times \mathrm{d}}$ respectively for nodes and predicates[3], where $d$ represents the embedding dimension, $N_{\mathrm{n}}$ represents the number of nodes, and $N_{\mathrm{p}}$ represents the number of predicate categories.

**Rewards**: During the path searching process, the agent only receives a terminal reward $R_{e,T}$ from the environment ($R_{e,t} = 0, \forall t < T$). Let $v_T$ be the last node of the path, and $\mathcal{N}_{drug}$ be the known

---

[3]Each sub-network uses separate embedding matrices.

diseases that drug $v_{drug}$ can treat, the terminal reward is calculated by using the proposed drug repurposing model (Sec. 4.1) via:

$$R_{e,T} = \begin{cases} 1, & \text{if } v_T \in \mathcal{N}_{drug}. \\ p_{treat}, & \text{if } v_T \notin \mathcal{N}_{drug}; v_T \in \mathcal{V}^{\text{disease}} \text{ and } f(v_{drug}, v_T) \text{ is predicted as "treat".} \\ 0, & \text{if } v_T \notin \mathcal{N}_{drug}; v_T \in \mathcal{V}^{\text{disease}} \text{ and } f(v_{drug}, v_T) \text{ is not predicted as "treat".} \\ -1, & \text{if } v_T \notin \mathcal{V}^{\text{disease}}. \end{cases}$$

where $p_{treat}$ is the "treat" class probability predicted by the drug repurposing model $f$.

The ADAC-based RL model consists of four sub-networks that share the same model architecture $\text{MLP}^i$ (note that $i$ represents the id of each sub-network described later, such as $a$ for *actor network*, $c$ for *critic network*, etc.) but with different parameters:

$$\text{MLP}^i(X) = \boldsymbol{f}(\boldsymbol{f}(\boldsymbol{f}(XW_1^i + b_1^i)W_2^i + b_2^i)W_3^i + b_3^i) \tag{2}$$

where $\{W_1^i, W_2^i, W_3^i, b_1^i, b_2^i, b_3^i\}$ are the parameters and biases of linear transformations, $\boldsymbol{f}$ represents a batch normalization layer followed by an ELU activation function.

**Actor network**: The actor network learns a path-finding policy $\pi_\theta$ (note that $\theta$ represents all parameters of actor network) to guide the agent to choose an action $a_t$ from the action space $A_t$ based on current state $s_t$:

$$\pi_\theta(a_t|s_t, A_t) = \text{softmax}(A_t \odot \text{MLP}^a(s_t)) \tag{3}$$

where $\odot$ represents the dot product. Here, $\pi_\theta(a_t|s_t, A_t)$ represents the probability of choosing action $a_t$ at time $t$ from the action space $A_t$ given the state $s_t$.

**Critic network**: The critic network (Lillicrap et al., 2015) estimates the expected reward $Q_\phi(s_t, a_t)$ (note that $\phi$ represents all parameters of critic network) if the agent takes the action $a_t$ at state $s_t$ by:

$$Q_\phi(s_t, a_t) = \text{MLP}^c(s_t) \odot a_t \tag{4}$$

**Path discriminator network**: Since the RL agent only receives a terminal reward indicating whether it reaches an expected target, to encourage the agent to find biologically reasonable paths and provide intermediate rewards, we further guide it with demonstration paths. This network is essentially a binary classifier that distinguishes whether a path segment $(s_t, a_t)$ is from demonstration paths or generated by the actor network. We treat all the known demonstration path segments $(s_t^D, a_t^D)$ as positive samples and all actor-generated non-demonstration path segments $(s_t, a_t)$ as negative samples. The path discriminator network $D_p(s_t, a_t) = \text{sigmoid}(\text{MLP}^p(s_t \oplus a_t))$, where $\oplus$ represents the concatenation operator, is optimized with:

$$L_p = -\mathbb{E}_{(s,a)\sim P_D}[\log(D_p(s,a))] - \mathbb{E}_{(s,a)\sim P_A}[\log(1 - D_p(s,a))] \tag{5}$$

where $P_D$ and $P_A$ respectively represent the demonstration path segment distribution and the actor-generated non-demonstration path segment distribution. Based on the probability $D_p(s_t, a_t)$, the path-discriminator-based intermediate reward $R_{p,t}$ is calculated as:

$$R_{p,t} = \log(D_p(s_t, a_t)) - \log(1 - D_p(s_t, a_t)). \tag{6}$$

**Meta-Path discriminator network**: Similar to the path discriminator, this network aims to judge whether the meta-path of the actor-generated paths is similar to that of demonstration paths. The meta-path is the path of node categories (e.g., ['Drug'→'Gene'→'BiologicalProcess'→'Disease']). Similarly, the meta-path discriminator $D_m(M) = \text{sigmoid}(\text{MLP}^m(M))$ is also a binary classifier where the meta-paths of demonstration paths are treated as positive samples while others are negative samples. We optimize it with the following loss:

$$L_m = -\mathbb{E}_{M\sim P_D^M}[\log(D_m(M))] - \mathbb{E}_{M\sim P_A^M}[\log(1 - D_m(M))] \tag{7}$$

where $M$ is a meta-path embedding vector defined as the concatenation of learned category embeddings of all nodes that appear in the path; $P_D^M$ and $P_A^M$ respectively represent the demonstration meta-path distribution and the actor-generated non-demonstration meta-path distribution. The intermediate reward $R_{m,t}$ generated by the meta-path discriminator is calculated by:

$$R_{m,t} = \log(D_m(M)) - \log(1 - D_m(M)). \tag{8}$$

The final intermediate reward $R_t$ at time $t$ is then calculated as:

$$R_t = \alpha_p R_{p,t} + \alpha_m R_{m,t} + (1 - \alpha_p - \alpha_m)\gamma^{T-t}R_{e,T} \tag{9}$$

where $\alpha_p \in [0, 1]$ and $\alpha_m \in [0, 1 - \alpha_p]$ are hyperparameters, $\gamma$ is the decay coefficient, and $R_{e,T}$ is defined in the "Rewards" section above. To optimize the critic network, we minimize the Temporal Difference (TD) error (Sutton, 1988) with loss:

$$L_c = \text{TD}^2 = [(R_t + Q_\phi(s_{t+1}, a_{t+1})) - Q_\phi(s_t, a_t)]^2. \tag{10}$$

Since the goal of the actor network is to achieve the largest expected reward by learning an optimal actor policy, we optimize the actor network by maximizing $J(\theta) = \mathbb{E}_{a \sim \pi_\theta}[Q_\phi(s_t, a)]$. We use the REINFORCE algorithm (Williams, 1992) to optimize the parameters. To encourage more diverse exploration in finding paths, we use the entropy of $\pi_\theta$ as a regularization term and optimize the actor network with the following stochastic gradient of the loss function $L_a$:

$$\nabla_\theta L_a = -\nabla_\theta J(\theta) = -\mathbb{E}_{\pi_\theta}[\nabla_\theta \text{TD} \log \pi_\theta(a_t|s_t)] - \alpha \nabla_\theta \text{entropy}(\pi_\theta) \tag{11}$$

where $\pi_\theta$ is the action probability distribution based on the actor policy and $\alpha$ is the entropy weight.

Following Zhao et al. (2020), we train the ADAC-based RL model in a multi-stage way. First, we initialized the actor network using the behavior cloning method (Pomerleau, 1991) in which the training set of demonstration paths is used to guide the sampling of the agent with Mean Square Error (MSE) loss. Then, in the first $z$ epochs, we freeze the parameters of the actor network and the critic network and respectively train the path discriminator network and meta-path discriminator network by minimizing $L_p$ and $L_m$. After $z$ epochs, we unfreeze the actor network and the critic network and optimize them together by minimizing a joint loss $L_{joint} = L_a + L_c$.

## 5 EXPERIMENTS

In this section, we introduce training data used in this study (Sec. 5.1), evaluation setup (Sec. 5.2) as well as evaluation results (Sec. 5.3 & 5.4) respectively for drug repurposing prediction and MOA prediction. Implementation details of the proposed models are presented in Appx. B.

### 5.1 TRAINING DATA

Training data of drug repurposing model consists of drug-disease pairs in three categories: "treat", "not treat", and "unknown". For the first two categories, we collect data from four different data sources (Appx. A.3.1). The raw data are pre-processed and separated into training, validation, and test sets with ratios of [0.8, 0.1, 0.1] for downstream model training (Appx. A.3.2).

### 5.2 EVALUATION METRICS AND METHODS

We evaluate the proposed models on two types of tasks: predicting drug-disease "treat" probability (i.e., drug repurposing prediction) and identifying biologically reasonable KG-based MOA paths from all candidates (i.e., MOA prediction). These two tasks are evaluated based on classification accuracy-based metrics (e.g., accuracy, macro f1 score) and ranking-based metrics (e.g., mean percentile rank (MPR), mean reciprocal rank (MRR), and Hit@K) defined as follows:

$$\text{MPR} = \frac{1}{|PR|} \sum_{pr \in PR} pr \qquad \text{MRR} = \frac{1}{|R|} \sum_{r \in R} r^{-1} \qquad \text{Hit@K} = \frac{1}{|R|} \sum_{r \in R} |r \le k|, \tag{12}$$

where $PR$ and $R$ are respectively a list of percentile ranks and a list of ranks of all true positive drug-disease pairs ("treat" category) in the evaluation dataset (combination of validation and test).

**Drug repurposing prediction**: Due to the extremely long running time of certain baselines (e.g., GraphSAGE+SVM), we calculate the metrics MRR and Hit@K based on the rank of each true positive drug-disease pair among 1000 random drug-disease pairs (500 with drug id replacement and 500 with disease id replacement) and compare them across all models. We also show the comparison results with MRR and Hit@K based on "all nodes" replacement for the models that can be implemented in a reasonable time (Appx. C.2). In addition, since our drug repurposing model does 3-class classification while other baselines do 2-class classification, for a fair comparison, we re-calculate "accuracy" and "macro f1 score" for our drug repurposing model by excluding the "unknown" class.

**MOA prediction**: For the evaluation of MOA prediction (Sec. 4.2), we use the DrugMechDB (Mayers et al., 2020), an expert-curated path-based drug MOA database, to obtain the verified MOA

paths as ground-truth data. We match each biological concepts in these verified MOA paths to the biological entities used in our customized knowledge graph (Sec. 3) and generate the corresponding KG-based MOA paths (Appx. A.4). The matched KG-based MOA path is defined as a 3-hop KG-based path of which all four nodes appear in the verified MOA paths. For each drug-disease pair not in the training set, we calculate the path scores for all 3-hop KG paths between drug and disease with the path-finding policy learned from the ADAC-based RL model by using equation: path score $= \sum_{i=1}^{k} \delta^{i-1} \times \log(P_i \times N_i)$, where $k$ is the number of hops in this path; $\delta$ is a decay coefficient (we set it to 0.9 in this study); $P_i$ represents the probability of choosing action $a_i$ in the $i$th hop following this path based on the trained ADAC-based RL model; $N_i$ is the number of possible actions in the $i$th hop. With these path scores, we obtain the ranks of the matched KG-based MOA paths and calculate the ranking-based metrics mentioned above. For those drug-disease pairs with multiple KG-based MOA paths, we use the highest ranks of their paths. We compare our ADAC-based RL model with the baseline models based on these metrics. In addition, we further perform some case studies to evaluate the effectiveness of the proposed method and present them in Appx. C.3.

### 5.3 DRUG REPURPOSING PREDICTION EVALUATION

**Baselines**: We compare our proposed drug repurposing prediction model against several state-of-the-art (SOTA) KG-based models and the variants of our proposed model. `TransE` (Bordes et al., 2013), `TransR` (Lin et al., 2015), `RotatE` (Sun et al., 2019) are the translation-distance-based models that regard a relation as a translation/rotation from a head entity to a tail entity. `DistMult` (Yang et al., 2014) is a bilinear model that measures the latent semantic similarity of a triple with a trilinear dot product. `ComplEx` (Trouillon et al., 2016) and `ANALOGY` (Liu et al., 2017) are the extensions of `DistMult` that consider more complex relations. `SimpLE` (Kazemi & Poole, 2018) is a tensor-factorization-based model to learn the semantic relation of a triple. `GAT` (Veličković et al., 2018) is a popular graph attention model. Besides these SOTA baselines, we use the variants of our proposed model (e.g., pure GraphSAGE for link prediction `GraphSAGE-link`, the `GraphSAGE+logistic` model, the `GraphSAGE+SVM` model, and the `GraphSAGE+RF` model for 2-class classification[4]). Implementation details of these baselines are presented in Appx. B.

**Evaluation results**: Table 1 shows the evaluation results of the proposed drug repurposing model and other baselines based on the metrics described in Sec. 5.2. As shown in the table, on the one hand, the proposed model outperforms almost all baselines in classification-based metrics even though it is a bit worse than the GAT model, which indicates its effectiveness in classifying known "treat" and "not treat" drug-disease pairs based on their attribute and neighborhood information on the knowledge graph (note that we conduct an ablation experiment to show the effectiveness of node attribute embeddings in Appx. C.1). On the other hand, the best performance of our model in ranking-based metrics shows its capability in identifying new indications of existing drugs out of all the possible drug-disease pairs with relatively low false positives, which is of great importance for guiding clinical research. Besides, comparing `2-class GraphSAGE+RF` with the vanilla GraphSAGE model (e.g., `GraphSAGE-link`), the results demonstrate the effectiveness of the Random Forest model over a neural network classifier in this task. Comparison between 2-class and 3-class GraphSAGE+RF models indicates the importance of using negative sampling to generate "unknown" drug-disease pairs for model training. With the "unknown" drug-disease pairs, the `3-class GraphSAGE+RF` model achieves significant improvement in ranking-based metrics, which is essential when applying to real-world drug repurposing because it can reduce the false positives.

### 5.4 MECHANISM OF ACTION (MOA) PREDICTION EVALUATION

**Baselines**: For a fair comparison, we choose the `MultiHop` model (Lin et al., 2018) (Implementation details in Appx. B) as a baseline since it allows using a self-defined reward shaping strategy in its reward function as what we do in our proposed ADAC-based RL model. We don't compare with other existing models that either require pre-known edge importance information (see Sec. 2) or cannot be trained within a reasonable time (e.g., within two weeks) on the customized knowledge graph. To show the importance of demonstration paths, we also compare with an ablated version of

---

[4] only considers true positive and true negative.

Table 1: Comparing evaluation results between the proposed drug repurposing model with baselines. The values with * inside the parenthesis are the results after excluding the "unknown" category.

| Model | Accuracy | Macro F1 score | MRR | Hit@1 | Hit@3 | Hit@5 |
|---|---|---|---|---|---|---|
| TransE | 0.706 | 0.706 | 0.275 | 0.111 | 0.295 | 0.446 |
| TransR | 0.858 | 0.855 | 0.307 | 0.130 | 0.350 | 0.517 |
| RotatE | 0.707 | 0.707 | 0.255 | 0.077 | 0.287 | 0.454 |
| DistMult | 0.555 | 0.495 | 0.172 | 0.040 | 0.144 | 0.258 |
| ComplEx | 0.624 | 0.456 | 0.133 | 0.023 | 0.106 | 0.196 |
| ANALOGY | 0.597 | 0.467 | 0.179 | 0.045 | 0.147 | 0.272 |
| SimplE | 0.600 | 0.475 | 0.161 | 0.037 | 0.135 | 0.236 |
| GAT | **0.932** | **0.929** | 0.002 | 0 | 0 | 0.001 |
| GraphSAGE-link | 0.919 | 0.915 | 0.002 | 0 | 0 | 0 |
| GraphSAGE+logistic | 0.787 | 0.779 | 0.002 | 0 | 0 | 0 |
| GraphSAGE+SVM | 0.809 | 0.795 | 0.002 | 0 | 0 | 0 |
| 2-class GraphSAGE+RF | 0.929 | 0.925 | 0.271 | 0.176 | 0.311 | 0.386 |
| 3-class GraphSAGE+RF (ours) | 0.934 (0.929*) | 0.922 (0.925*) | **0.360** | **0.211** | **0.410** | **0.524** |

Table 2: Evaluation results on MOA prediction on MultiHop model and ADAC-based RL model with and without demonstration paths (ADAC RL w/o DP).

| Model | MPR | MRR | Hit@1 | Hit@10 | Hit@50 | Hit@100 | Hit@500 |
|---|---|---|---|---|---|---|---|
| MultiHop | 61.541% | 0.026 | 0.012 | 0.043 | 0.081 | 0.124 | 0.360 |
| ADAC RL w/o DP | 73.281% | 0.022 | 0.012 | 0.025 | 0.112 | 0.186 | 0.447 |
| ADAC RL w/ DP (ours) | **94.410%** | **0.123** | **0.062** | **0.242** | **0.509** | **0.640** | **0.857** |

the proposed model that does not take advantage of the demonstration paths by setting $\alpha_p$ and $\alpha_m$ in Function 9 as 0.

**Evaluation results**: Table 2 shows the evaluation results. Although all the models receive the same terminal rewards from the environment, the proposed ADAC-based RL model achieves significantly better performance in identifying biologically reasonable KG-based MOA paths than other two baselines. Comparison between the proposed model and the ADAD-based RL model without demonstration paths (i.e., ADAC RL w/o DP) further illustrates the great effectiveness in using demonstration paths to guide the path-finding process. Due to the massive searching space and sparse rewards, the RL agent often fails to find reasonable paths out of plenty of possible choices, while our model, with the intermediate guidance provided by the demonstration path, is able to identify the most biologically reasonable choices at each time step with a much higher probability. Moreover, we perform two specific case studies (see Appx. C.3) where we further show the top-rank MOA paths predicted by our model are able to identify key molecules in real drug action regulatory networks.

## 6 CONCLUSION

In this paper, we propose a computational drug repurposing model framework that predicts not only the treatment probabilities of drug-disease pairs but also the KG-based mechanism of action (MOA) paths as biological explanations for drug repurposing. We apply the proposed model framework to a large and standardized biomedical knowledge graph and conduct experiments to compare our framework with extensive state-of-the-art KG-based models. The results show that our proposed drug repurposing model achieves much better predictive performance in a comprehensive evaluation. We also show that by leveraging the proposed knowledge-and-publication-based demonstration paths to provide intermediate guidance during the path-finding process, our model can effectively find biologically reasonable paths out of a large amount of possible choices. Furthermore, two specific case studies are performed to illustrate the effectiveness of our models in identifying some key biological molecules in real drug action regulatory networks. We believe our proposed framework can effectively reduce the "black-box" concerns and increase prediction confidence for drug repurposing based on its predicted path-based explanations, which further accelerates the process of drug discovery for emerging diseases. Despite the comparably good performance of our models, some limitations remain to be solved. For example, our MOA prediction is based on an assumption that a repurposed drug follows similar molecular mechanisms as the known MOAs to treat different diseases, which needs further biologically experimental verification.

## 7 ETHICS STATEMENT

Our proposed method aims to provide a new model framework for drug repurposing prediction and explanation. The potential negative societal impacts might include the risk of misusing the drug repurposing capability (e.g., misleading patients to take drugs inappropriately). Although our model can provide certain suggestions for the new indications of drugs, it is mainly designed for assisting medical practitioners (e.g., doctors and licensed practitioners), who have professional training or knowledge to assess the accuracy of suggestions provided by our model. Therefore, the application of this method should be limited to drug development in pharmaceutical companies or pre-clinical drug research in relevant academic or medical organizations, but not be directly accessible to the public. Since drugs treat diseases in complex ways, there is still much unknown knowledge involved, and therefore any suggestions given by this proposed method should require further clinical testing but not be used to directly diagnose, prevent, or treat any diseases without FDA approval. In addition, although the data used in this study (including the data in the biomedical knowledge graph *RTX-KG2c*) are from multiple data sources, they are all collected from publicly accessible databases or datasets or APIs with free licenses (except for DrugBank data that we have requested and obtained its non-commercial license). Note that the licenses of all data are shown in Table 3. Therefore, they have excluded any personal or sensitive information. All code used in this study is originally created by ourselves or modified from the code of previous researchers (note that we illustrate this at the top of those scripts), and allowed to be disclosed with the permission of its original authors.

Table 3: Summary of licenses of all data we used.

| Existing assets | Licenses |
|---|---|
| RTX-KG2c (v2.7.3) https://github.com/RTXteam/RTX-KG2 | MIT license, CC-BY 4.0 license |
| DrugMechDB data (v1.1.0) https://github.com/SuLab/DrugMechDB | MIT license |
| DrugBank data (v5.1) https://go.drugbank.com | DrugBank academic license |
| Molecular Data Provider (v1.2) https://github.com/broadinstitute/molecular-data-provider | MIT license |
| MyChem data (v1) https://mychem.info | Apache-2.0 license |
| SemMedDB data (semmedVER43_R) https://lhncbc.nlm.nih.gov/ii/tools/SemRep_SemMedDB_SKR.html | UMLS Metathesaurus license |
| NDF-RT data (v2018AA) https://bioportal.bioontology.org/ontologies/NDFRT | UMLS Metathesaurus license |
| RepoDB data https://unmtid-shinyapps.net/shiny/repodb | CC-BY 4.0 license |

## 8 REPRODUCIBILITY STATEMENT

We share our code via the following link `https://drive.google.com/file/d/1AnwkHKZ69d_9twfLy1fWNn9Ssey7YYxj/view?usp=sharing`. The code includes the steps to download our customized knowledge graph and other required datasets (e.g., DrugBank data. Please note that to download DrugBank data, the DrugBank non-commercial license has to be requested and obtained by following the instructions on the DrugBank website), pre-process the raw data, train different models (e.g., our proposed models and baseline models), and evaluate models. The sources of all datasets used in this study are listed in Table 3 and the complete description of these datasets with their pre-processing steps is provided in Appx. A. The implementation details of all models can be found in Appx. B.

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

# A   DATA COLLECTION AND PRE-PROCESSING

## A.1   BIOMEDICAL KNOWLEDGE GRAPH RTX-KG2C PRE-PROCESSING

The dataset of *RTX-KG2c* (v2.7.3) (Wood et al., 2022) is accessed via https://github.com/RTXteam/RTX-KG2, which contains around 6.4M concept nodes with 56 distinct categories and 39.3M relationship edges with 77 distinct relations. We pre-process the raw data of *RTX-KG2c* by the following four principles:

1. Since we are mainly interested in the categories relevant to drug mechanisms of action (MOAs), we exclude the nodes with categories that are not expected to be useful for drug re-purposing explanation (e.g., "GeographicaLocation", "Device", "InformationResource").

2. One of data sources used in the *RTX-KG2c* is the Semantic MEDLINE Database (SemMedDB) (Kilicoglu et al., 2012), one of the most widely used NLP-derived biomedical knowledge sources, that has been found recently (Cong et al., 2018) to contain many noisy relations due to the immature NLP techniques even though it contains many latest-found relations (e.g., the relations with Covid-19). To improve the quality of SemMedDB-based edges, we filter out parts of SemMedDB edges based on the criteria that each remaining SemMedDB edge has to be supported by at least 10 publications as well as the PubMed-publication-based NGD score (Cilibrasi & Vitanyi, 2007) (described in Appx. A.2) of its two end nodes should not be higher than 0.6

3. The *RTX-KG2c* is a multigraph that allows multiple edges connecting between two nodes. The edge relations (a.k.a. predicates) follow the predicate hierarchy [5] used in the Biolink model (Unni et al., 2022). Therefore, the *RTX-KG2c* contains some hierarchically redundant edges between two nodes. To reduce the complexity of path finding for the downstream MOA predictions, we only reserve the "leaf" predicates of the Biolink semantic relation hierarchy if there exists hierarchically associated edges between two nodes. For example, if the edges between two nodes are "affected by", "entity regulated by entity" and "entity positively regulated by entity", we remove the "affected by" and "entity regulated by entity" edges because they are the "ancestor" predicates of "entity positively regulated by entity" based on the predicate hierarchy. Removing those "ancestor" predicates does not affect the path explanation because the "leaf" predicates contain more precise semantic information than their "ancestor" predicates.

4. To prevent the training information leakage, we exclude all existing edges connecting between the potential drug nodes (the nodes with the categories of "Drug" and "Small-Molecule") and the potential disease nodes (the nodes with the categories of "Disease", "PhenotypicFeature", "BehavioralFeature" and "DiseaseOrPhenotypicFeature") in the RTX-KG2c.

After the above pre-processing steps, 3,659,165 nodes with 33 distinct categories (Figure 3 a) and 18,291,237 edges with 74 distinct types (Figure 3 b) are left for our customized *RTX-KG2c*.

## A.2   DEMONSTRATION PATH EXTRACTION

The demonstration paths are a set of multi-hop KG-based paths used to guide the agent in the ADAC-based RL model to find biologically reasonable KG-based MOA paths. It can be formulated as $P^k = \{p_{s,t}^k | v_s \in \mathcal{V}^{\text{drug}}; v_t \in \mathcal{V}^{\text{disease}}\}$ where $p_{s,t}^k$ is a multi-hop demonstration path with maximum path length $k$ starting from a drug node $v_s$ and ending at a disease node $v_t$. Given a drug node and a disease node (defined in Appx. A.1), the number of paths in the knowledge graph between them increases exponentially when $k$ increases. Therefore, we set $k = 3$ to guarantee that the agent can find the biologically meaningful predicted paths in a reasonable amount of time. We extract the reasonable demonstration paths from the customized *RTX-KG2c* (defined in Sec.3) by using the known drug-target interactions collected from two curated biomedical data sources (e.g., DrugBank (v5.1) and Molecular Data Provider (v1.2)) and an adjusted PubMed-based version of Normalized Google Distance (NGD) score (Cilibrasi & Vitanyi, 2007) formulated as follows:

---

[5]Visualization of hierarchical predicates in the Biolink Model v2.1.0 `http://tree-viz-biolink.herokuapp.com/predicates/2.1.0`

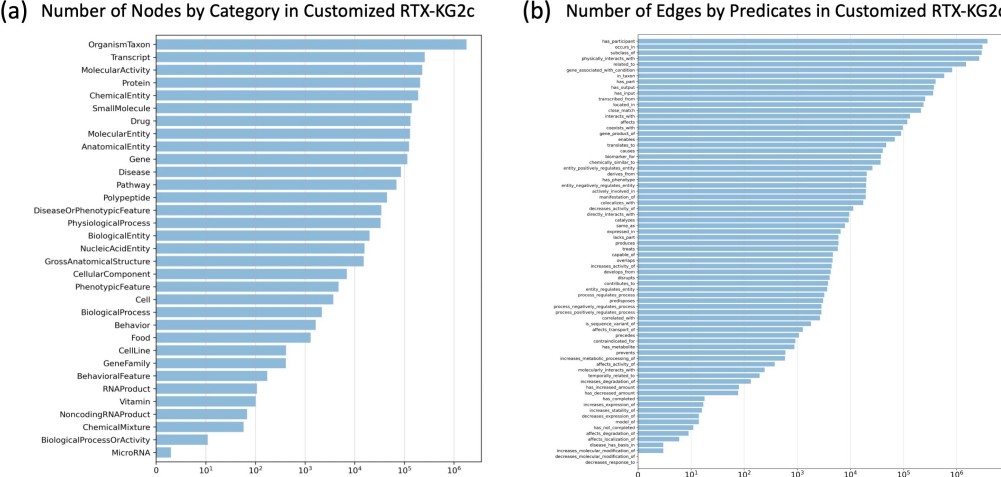

Figure 3: Number of nodes by category (a) and number of edges by predicate (b) in customized *RTX-KG2c*.

$$NGD(c1, c2) = \frac{max\{logf(c1), logf(c2)\} - logf(c1, c2)}{logN - min\{logf(c1), logf(c2)\}} \tag{13}$$

where $c1$ and $c2$ are two biological concepts used in the customized *RTX-KG2c* (defined in Sec.3); $f(c1)$ and $f(c2)$ respectively represent the total number of unique PubMed IDs associated with $c1$ and $c2$; $f(c1, c2)$ is the total number of unique PubMed IDs shared between $c1$ and $c2$ (All associated PubMed IDs are stored as node attributes in our customized *RTX-KG2c*); $N$ is the total number of pairs of Medical Subject Heading (MeSH) terms annotations in PubMed database[6].

By using both curated drug-target interactions and PubMed-publication-based NGD scores, the edges within a demonstration path must satisfy the following two requirements: 1. the edge between the drug node and the first intermediate node needs to be confirmed by the DrugBank (v5.1) or the Molecular Data Provider (v1.2) as well as has a NGD score that is not higher than 0.6; 2. the edge between the second intermediate node and the disease node should have a NGD score that is not higher than 0.6. According to these requirements, we finally find 396,705 demonstration paths for 8,495 true positive drug-disease pairs (described in Appx. A.3.2) and we split them into training, validation, and test sets for ADAC-based RL model training.

## A.3 TRAINING DATA

### A.3.1 DATA COLLECTION

We collect data from four different data sources for training our purposed models:

**MyChem Data** (Xin et al., 2018) is provided by the BioThing API collection (Xin et al., 2016), which contains up-to-date annotations regarding indication and contraindication for drugs.

**SemMedDB Data** (Kilicoglu et al., 2012) is provided by the Semantic MEDLINE Database (SemMedDB) which leverages natural language processing (NLP) techniques to extract semantic triples with "treats" and "negatively treats" relations from PubMed abstracts.

**NDF-RT Data** (Brown et al., 2004) is provided by National Drug File – Reference Terminology from Veterans Health Administration (VHA) which contains information on drug interaction, indications, and contraindications.

---

[6]PubMed (https://arax.ncats.io/) is a MEDLINE database that contains abundance references and abstracts on life sciences and biomedical topics

Table 4: Pair count of true positive (indications) and true negative (contraindications or no effect) data from four data sources (Appx. A.3.1). Note that 'shared' means those pairs are from two or more data sources.

| Source | True Positive (Treats) | True Negative (Not Treat) |
|---|---|---|
| MyChem | 3,663 | 26,795 |
| SemMedDB | 8,255 | 11 |
| NDF-RT | 3,421 | 5,119 |
| RepoDB | 2,127 | 738 |
| Shared | 3,971 | 526 |
| **Total** | 21,437 | 33,189 |

**RepoDB Data** (Brown & Patel, 2017) is a standard set of successful and failed drug-disease pairs in clinical trials collected by the Blavatnik Institute at Harvard Medical School.

### A.3.2 Pre-processing

To ensure our proposed models can be trained on high-quality data, we pre-process the raw data collected from four data sources (see Appx. A.3.1) by the following procedures:

1. We match the original identifiers of drugs and diseases from these four data sources to the identifiers used in the customized *RTX-KG2c* (defined in Sec.3), and then we remove duplicate drug-disease pairs and any pairs that appear in both true positives (pairs with relation "indication" from **MyChem Data**, predicate "treats" from **SemMedDB Data**, therapeutics "indications" from **NDF-RT Data** and status "approved" from **RepoDB Data**) and true negatives (pairs with relation "contraindication" from **MyChem Data**, predicate "neg_treats" from **SemMedDB Data**, therapeutics "contraindications" from **NDF-RT Data** and status "withdrawn" from **RepoDB Data**).

2. Due to publication bias and possible NLP mistakes that exist in **SemMedDB Data** (Cong et al., 2018), for those drug-disease pairs derived from this source, we select only those with support of at least ten publication abstracts for both true positive and true negative pairs. In addition, to increase the quality of SemMedDB-based data, we further filter the drug-disease pairs by using the PubMed-publication-based Normalized Google Distance (NGD) (Cilibrasi & Vitanyi, 2007) scores with a cutoff of 0.6 (described in Appx. A.2).

Table 4 shows the drug-disease pair count from each data source after data pre-processing.

For the drug repurposing model (Sec. 4.1) training, since we train a trinary classifier, we generate an "unknown" class by negative sampling, that is, replacing the drug or disease identifier in each true positive drug-disease pair with a random drug or disease identifier to generate a new pair that does not appear in both the "treat" and "not treat" classes. Specifically, for each unique true positive drug-disease pair, we respectively replace its drug identifier with other 30 random drug identifiers as well as replace its disease identifier with other 30 random disease identifiers to make 60 new drug-disease pairs for the "unknown" class.

For the ADAC-based RL model (Sec. 4.2.2) training, only the true positive pairs are considered. In addition, we remove those true positive pairs which are not reachable from a drug node to its corresponding disease node within maximum of 3 hops in the customized *RTX-KG2c* and this results in 19,332 true positive pairs being left. Out of these 19,332 true positive pairs, 8,495 pairs can find at least one demonstration path using the method described in Appx. A.2.

We split the data into training, validation, and test sets based on the ratio of [0.8, 0.1, 0.1]. Table 5 summarizes the pair count used as training, validation, and test sets for the drug repurposing model and ADAC-based RL model.

### A.4 DrugMechDB MOA path mapping

We access the DrugMechDB (Mayers et al., 2020) data via https://github.com/SuLab/DrugMechDB which contains 3,593 MOA paths for 3,327 unique drug-disease pairs. These paths are extracted

Table 5: Pair count of true positive (indications) pairs, true negative (contraindications or no effect) pairs, and random pairs respectively in training set, validation set, and test set for the drug repurposing model and ADAC-based RL model.

| Data | Drug Repurposing Model | | | Adversarial RL Model |
|---|---|---|---|---|
| | TP | TN | Random | TP |
| Training Set | 17,149 | 26,552 | 34,306 | 6,796 |
| Validation Set | 2,143 | 3,318 | 4,286 | 849 |
| Test Set | 2,145 | 3,319 | 4,290 | 850 |
| **Total** | 21,437 | 33,189 | 42,882 | 8,495 |

from free-text descriptions from DrugBank, Wikipedia, and other literature sources and then have been curated by subject experts. For these 3,327 unique drug-disease pairs, we first match them to the biological entities used in our customized *RTX-KG2c* (defined in Sec.3), and then we filter out those drug-disease pairs whose MOA paths don't have intermediate nodes. Finally, 2,893 pairs out of 3,327 pairs are reserved after filtering. Since these MOA paths are extracted from the human free-text description, the length of these MOA paths is varying. Based on a previous report (Mayers et al., 2022), there are many edges/relations of these MOA paths that are missing in the existing biomedical databases and thus it is difficult to match the complete MOA paths to the KG-based paths in our customized biomedical knowledge graph. In addition, since the length of predicted paths generated by our model is fixed to 3, we consider those 3-hop KG-based paths of which all four nodes show up in the complete MOA paths are the correct matched paths. Based on this definition of correct matched paths, we finally find 472 unique drug-disease pairs of which each has at least one such correct matched path in its all possible 3-hop paths between drug and disease in our customized biomedical knowledge graph. For those 2,421 unique drug-disease pairs that are filtered out because of no such correct matched path, most of them are due to the missing edges in our customized biomedical knowledge graph which is consistent with the findings in the previous analysis (Mayers et al., 2022). There are two reasons that could be used to explain these missing edges: 1) we filter out many low-quality edges from SemMedDB (described in Appx. A.1); 2) RTX-KG2c might miss some data sources that support these missing edges.

## B    Implementation details

### B.1    Drug repurposing model (Sec. 4.1) and its baselines (Sec. 5.3)

We utilize the source code [7] provided by the GraphSAGE paper (Hamilton et al., 2017) to train the unsupervised GraphSAGE embeddings with its "big" mean-based aggregator and two hidden layers with dimensions [256, 256]. In the random walk setting in GraphSAGE, we use 10 numbers of walk with 100 walk length. As for other parameters, the number of epochs is 10, the neighbor sampling size of each layer is 96, the learning rate is set to 0.001, the batch size is 256 and the maximum number of iterations per epoch is 10,000. Instead of using the default identity embeddings as the initial embedding features, we use node attribute embeddings generated by the pre-trained PubMedBert model (Gu et al., 2022) with concatenation of the node's name and category, and then are reduced to 100 dimensions with PCA technique. The dimension of final output embedding vector of each node is 512. With these GraphSAGE embedding vectors, we then concatenate the embedding vectors of each drug-disease pair in the training set as input features and use the *RandomForestClassifier* function of scikit-learn (v1.0) python package to train a Random Forest model. We run a grid search via *GridSearchCV* function to determine the optimal parameter set of Random Forest model from depths {5, 10, 15, 20, 25, 30, 35} and number of trees {500, 1000, 1500, 2000}. The best parameter set of the Random Forest model uses the maximum depth $max\_depth = 35$ and the number of trees $n\_estimators = 2000$.

For KG-based baseline models, we use the OpenKE library [8] to implement the TransE, TransR, RotatE, DistMult, ComplEx, ANALOGY, and SimpLE models with their default parame-

---

[7] https://github.com/williamleif/GraphSAGE
[8] https://github.com/thunlp/OpenKE

Table 6: Hyperparameters used for Baseline Models.

| Model | Hidden Dim. | Num. Epochs | Batch Size | Learning Rate | Optimizer |
|---|---|---|---|---|---|
| TransE | 100 | 10000 | 1000 | 1 | SGD |
| TransR | 50 | 2000 | 1000 | 1 | SGD |
| RotatE | 30 | 2000 | 1000 | 2e-5 | Adam |
| DistMult | 100 | 10000 | 1000 | 0.5 | Adagrad |
| ComplEx | 50 | 2000 | 500 | 0.5 | Adagrad |
| ANALOGY | 20 | 2000 | 500 | 0.5 | Adagrad |
| SimpLE | 100 | 2000 | 500 | 0.5 | Adagrad |

ter settings (we adjusted the hyperparameters for some models to ensure we can finish the training on GPUs for a reasonable time). Table 6 shows the detailed hyperparameter setting of each OpenKE-based baseline model. For other baseline models, we use the PyTorch Geometric [9] to implement the `GAT` and `GraphSAGE-link` model and use the same GraphSAGE embeddings (mentioned above) with scikit-learn (v1.0) python package to implement the `GraphSAGE+logistic`, `GraphSAGE+SVM` and `2-class GraphSAGE+RF` models with the grid-search-based optimal parameter settings.

## B.2 ADAC-BASED RL MODEL (SEC. 4.2.2) AND ITS BASELINE (SEC. 5.4)

We modify the model framework of Zhao et al. (Zhao et al., 2020) and make it adapted to the proposed methods (Sec. 4.2) for drug repurposing purpose. We respectively set the state history length $K = 2$ and the maximum length of path $T = 3$. In order to make our customized knowledge graph enable training on a 48GB Quadro RTX 8000 GPU, we prune the action space of each entity with a maximum size of 3,000 based on the PageRank score. We set 100 to the dimensions of all lookup matrices used within actor, critic, meta-path discriminator, and path discriminator networks. The dimensions of hidden layers of the actor network and critic network are all set to 512. We set the dimensions of hidden layers of the path discriminator with [512, 512] and used the dimension set of [512, 256] for the hidden layers of the meta-path discriminator. We utilize Xavier initialization (Glorot & Bengio, 2010) for the embeddings of all lookup matrices and the network layers. We set $\alpha_p = 0.006$ for the weight of the path discriminator reward and $a_m = 0.012$ for the meta-factor of the path discriminator reward. The decaying coefficient $\gamma$ of $R_{e,T}$ and the weight $\alpha$ of entropy term are respectively assigned 0.99 and 0.005. We optimize all networks with the Adam optimization algorithm (Kingma & Ba, 2015) with a learning rate of 0.0005. The mini-batch size is set to 32 with a path rollout set to 35. The dropout rates of all subnetworks are set to 0.3 and the action dropout rate is set to 0.5.

For the implementation of the `MultiHop` model (Lin et al., 2018), we use the source code[10] provided by the author and modify it by using our defined reward shaping strategy (described in Sec. 4.2.2). We set all its parameters the same as the ADAC-based RL model if they are available otherwise we use the default parameters.

## C OTHER EXPERIMENT RESULTS

### C.1 ABLATION EXPERIMENT FOR NODE ATTRIBUTE EMBEDDINGS

To show the effectiveness of node attribute embeddings (mentioned in Sec. 4.1) in improving repurposing predictions, we conduct an ablation experiment to compare the 3-class GraphSAGE+RF model with and without node attribute embeddings, using random embeddings with Xavier initialization to replace node attribute embeddings in the latter case. The results in Table 7 shows that the GraphSAGE initialized with node attribute embedding result in the final node embeddings with more expressive power and can help improve the performance of drug repurposing prediction.

---

[9]`https://github.com/pyg-team/pytorch_geometric`
[10]`https://github.com/salesforce/MultiHopKG`

Table 7: Comparing evaluation results between the 3-class GraphSAGE+RF model with and without node attribute embeddings

| Model | Accuracy | Macro F1 score | MRR | Hit@1 | Hit@3 | Hit@5 |
|---|---|---|---|---|---|---|
| 3-class GraphSAGE+RF w/o node attribute embedding | 0.907 | 0.889 | 0.151 | 0.031 | 0.134 | 0.239 |
| 3-class GraphSAGE+RF w/ node attribute embedding | **0.934** | **0.922** | **0.360** | **0.211** | **0.410** | **0.524** |

## C.2 Drug repurposing prediction evaluation with "all nodes" replacement

To further demonstrate that our proposed drug repurposing model has better capability in prediction with relatively low false positive, besides showing the comparison results based on 1000 random drug-disease pairs across all models (see Table 1), we calculate the MRR and Hit@K metrics for the models that can be implemented in a reasonable time (excluding the `GAT` and `GraphSage+SVM` models) by using the following three methods to replace drugs and/or diseases in each true positive drug-disease pair ("treat" category) in the evaluation dataset (combination of validation and test sets):

- **Drug-rank-based Replacement**: For each true positive drug-disease pair, the drug-rank-based replacement pairs are generated by replacing the drug entity with each of all 274,676 other drugs in the customized *RTX-KG2c* while excluding all known true positive drug-disease pairs.

- **Disease-rank-based Replacement**: For each true positive drug-disease pair, the disease-rank-based replacement pairs are generated by replacing the disease entity with each of all 124,638 other diseases in the *RTX-KG2c* while excluding all known true positive drug-disease pairs.

- **Combined Replacement**: For each true positive drug-disease pair, the combined replacement pairs are the combination of all replacement pairs of the above two methods. All known true positive drug-disease pairs are excluded from these replacement pairs.

Figure 4 shows the comparison results of MRR and Hit@K between our proposed drug repurposing model (i.e., the `3-class GraphSage+RF` model) and other baselines (excluding the `GAT` and `GraphSage+SVM` models because of extremely long running time) by using the above three methods to generate drug-disease replacement pairs. These results based on the ranking-based metrics calculated with more comprehensive "all nodes" replacement methods can indicate our proposed model is indeed superior to other baselines in identifying new indications of existing drugs with relatively low false positives.

## C.3 Case studies

To further evaluate the performance of the ADAC-based RL model in predicting MOA paths for drug repurposing explanation, we utilize it to explore potential repurposed drugs for the orphan or rare genetic diseases (e.g., Hemophilia B and Huntington's Disease) and predict their top 10 KG-based 3-hop paths as explanations. These two diseases have known molecular pathogenesis which can be used to evaluate if the predicted paths are biologically reasonable as an explanation for drug repurposing.

### C.3.1 Hemophilia B

Hemophilia B (a.k.a factor IX deficiency or Christmas disease) is a rare genetic disease that can cause longer-than-normal bleeding in patients. It is caused by the mutations in the factor IX (F9) gene on the X chromosome. Table 8 displays the top 10 predicted drugs/treatments from the proposed drug repurposing model (Sec. 4.1) which include both drugs/treatments used in the training set (highlighted with red color) and drugs/treatments not in the training set. We manually exclude the chemotherapeutic drugs from the predicted drug candidate list due to their potential risk of cytotoxicity to normal cells, which might lead to false positives for drug repurposing of non-cancer diseases (Sourimant et al., 2021; Gysi et al., 2021). For those seven drugs/treatments that are not in the training set, many of them are biologically reasonable and supported by some publications that have the potential to treat hemophilia B even though there are three predicted drugs that have

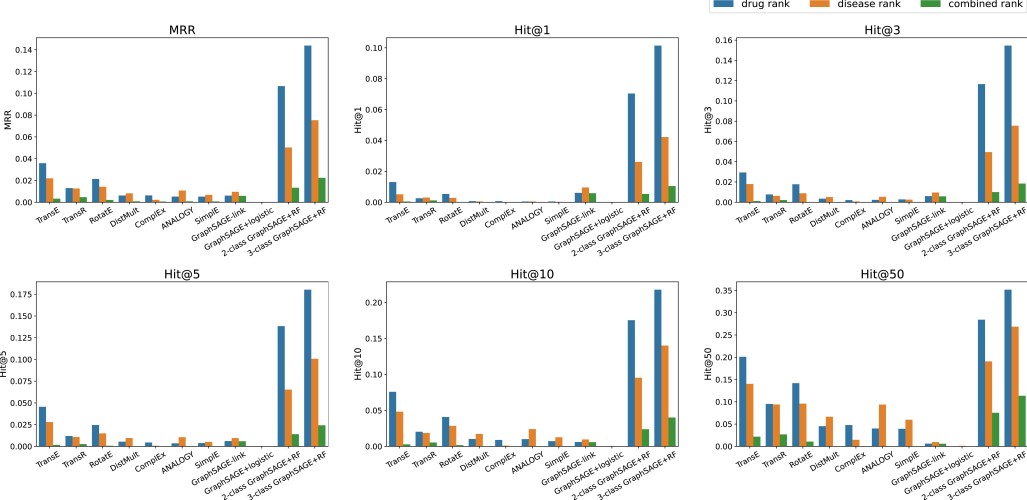

Figure 4: Comparison results of MRR and Hit@K between our proposed drug repurposing model (i.e., the `3-class GraphSage+RF` model) and other baselines (excluding `GAT` and `GraphSage+SVM` models) based on the evaluation dataset (combination of validation set and test set). The legend "drug rank", "disease rank" and "combined rank" respectively correspond to the methods of "Drug-rank-based Replacement", "Disease-rank-based Replacement", and "Combined Replacement" described in Appx. C.2.

no supporting publication. By the predicted 3-hop KG-based MOA paths, the domain experts might be able to quickly determine whether they are false positive drugs. For instance, Epicriptine is a nootropic drug that has no 3-hop KG-based path connecting this drug and hemophilia B and thus might be a false positive. Hyperbaric Oxygen therapy is reported to have a function in wound healing in which the first phase is to stop bleeding (Huang et al., 2019) and there is only one 3-hop KG-based path ("Hyperbaric oxygen" → "entity positively regulates entity" → "Angiogenic process" → "has participant" → "Epidermal growth factor" → "gene associated with condition" → "Hemophilia B") connecting between hyperbaric oxygen and hemophilia B, which might help experts to determine if it can be used as adjunctive therapy for the treatment of hemophilia B. Triamcinolone is a kind of steroids which might be beneficial for the adjunctive treatment of hemophilia (Cacciotti et al., 2021). Even though steroids such as triamcinolone might typically be used topically, a therapeutic steroid might be used orally. Therefore, it is not surprising that triamcinolone is returned as a possible treatment. In its predicted 3-hop KG-based MOA paths (see Figure 5), some genes like the AR gene (MORGAN et al., 1997) and the CD4 gene (Dobrzynski et al., 2004) are reported to be associated with the regulatory mechanism of hemophilia B. So, these might help experts further determine if triamcinolone is useful for hemophilia B.

In order to further evaluate the biological explanations of our predicted 3-hop KG-based MOA paths, we utilize the curated DrugMechDB-based MOA paths. There are three relevant MOA paths found for hemophilia B respectively corresponding to Eptacog Alfa, Eftrenonacog Alfa, and Nonacog Alfa. Eptacog Alfa and Nonacog Alfa are in our top 10 predicted drugs/treatments which are all used in the training set. Figure 6 shows the comparisons between the subgraphs with the top 10 predicted 3-hop KG-based paths and the curated DugMechDB-based MOA paths for Eptacog Alfa and Nonacog Alfa. The corresponding biological entities between the predicted paths and the curated DrugMechDB-based paths are highlighted with red color. Although the predicted paths can't exactly match the DrugMechDB-based MOA paths due to the limited path length and some missing semantic relationships, some key biological entities (e.g., Coagulation Factor VII, Coagulation Factor X, and Coagulation Factor IX) for the treatment of hemophilia B show up in the subgraph of top10 predicted paths. Actually, the treatment of hemophilia B involves a complex molecular network of blood coagulation (see Figure 7). Many biological entities (e.g., Coagulation factor VII, factor VIIa, factor III, factor II, factor VIII, factor IX, factor X) in the subgraph of our top 10 predicted paths are

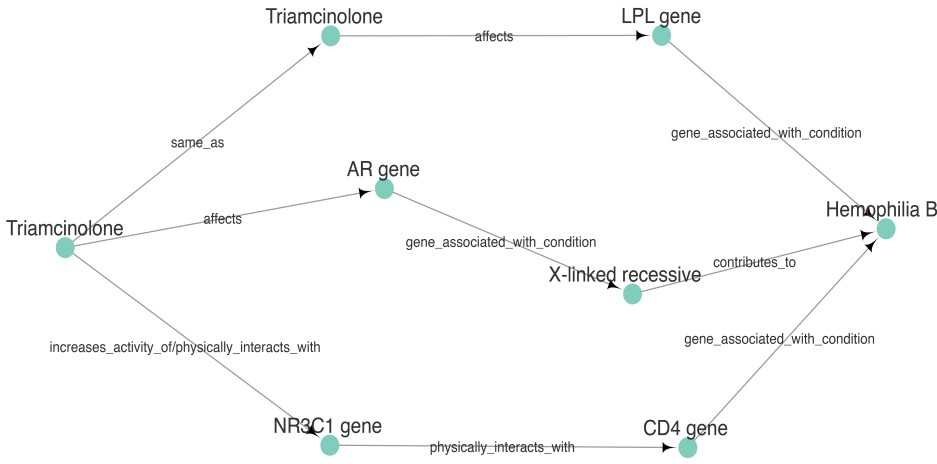

Figure 5: The predicted 3-hop KG-based MOA paths for triamcinolone.

in such molecular network. Therefore, the subgraph constructed by our top 10 predicted paths can convey more molecular details than the DrugMechDB-based MOA paths.

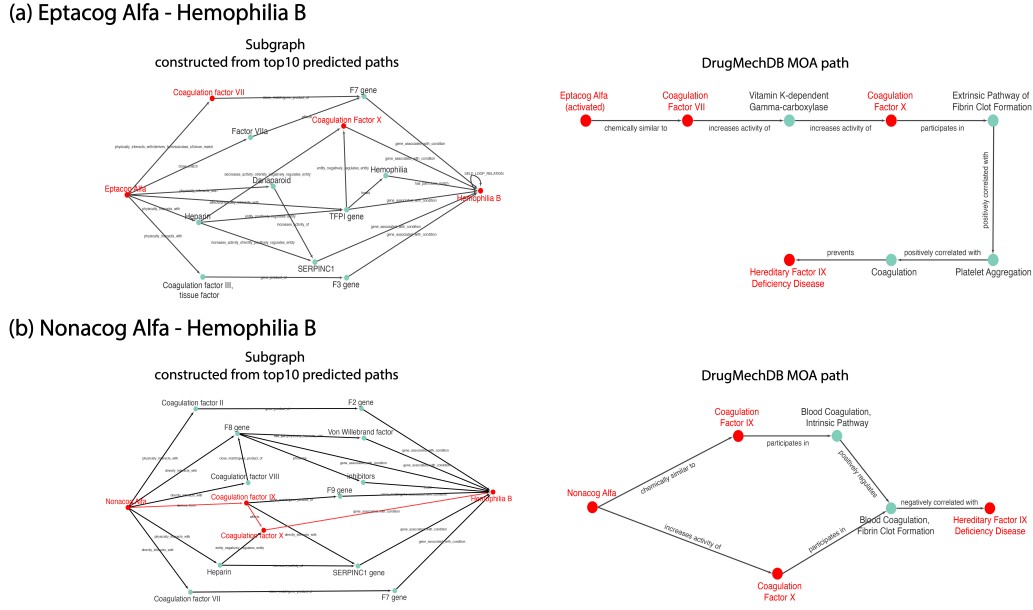

Figure 6: Comparison between the top 10 predicted 3-hop paths (integrated into a subgraph for better visualization) and the curated DugMechDB-based MOA paths (the original MOA paths are presented in the same way as the predicted paths) for Eptacog Alfa (a) and Nonacog Alfa (b). The corresponding biological entities between top10 predicted 3-hop paths and MOA paths are highlighted with red color. The top 10 predicted paths where all entities show up in the DugMechDB-based MOA paths are also highlighted with red color.

Since both Eptacog Alfa and Nonacog Alfa are used in the training set, to evaluate the predicted MOA paths for the predicted drugs/treatments that are not in the training set, we use Factor VIIa which has the highest probability after excluding all drugs used in the training set. Figure 8 shows a sbugraph with its top10 predicted 3-hop paths. We can see that most biological entities (e.g., Thrombin, Factor IX, factor VII) are biologically relevant (which are in the blood coagulation regulatory network). The predicted MOA paths (such as "Factor VIIa" → "entity positively regulates entity" → "Factor IX" → "close match" → "F9 (Coagulation Factor IX) gene" → "gene associated with

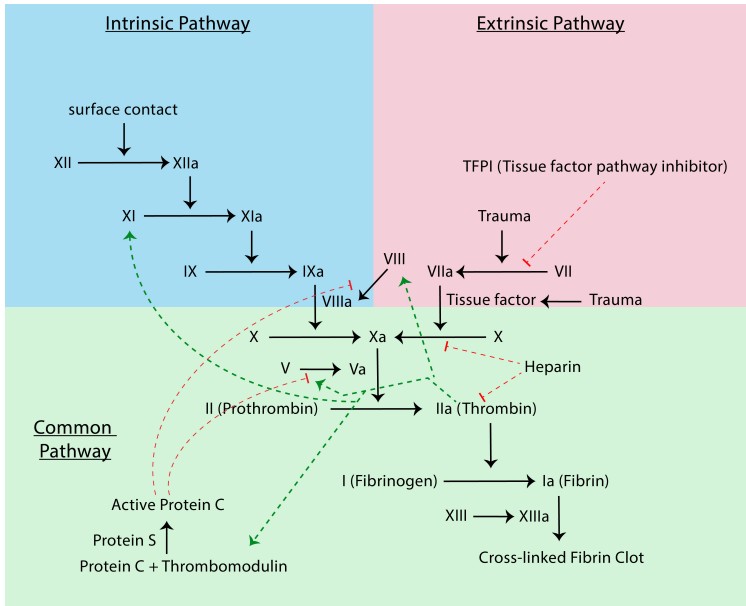

Figure 7: Blood Coagulation Regulatory Network with arrows for molecular reactions (black), positive feedback (green), and negative feedback (red).

condition" → "Hemophilia B") are biologically reasonable because the pathogenesis of hemophilia B is the deficiency of factor IX.

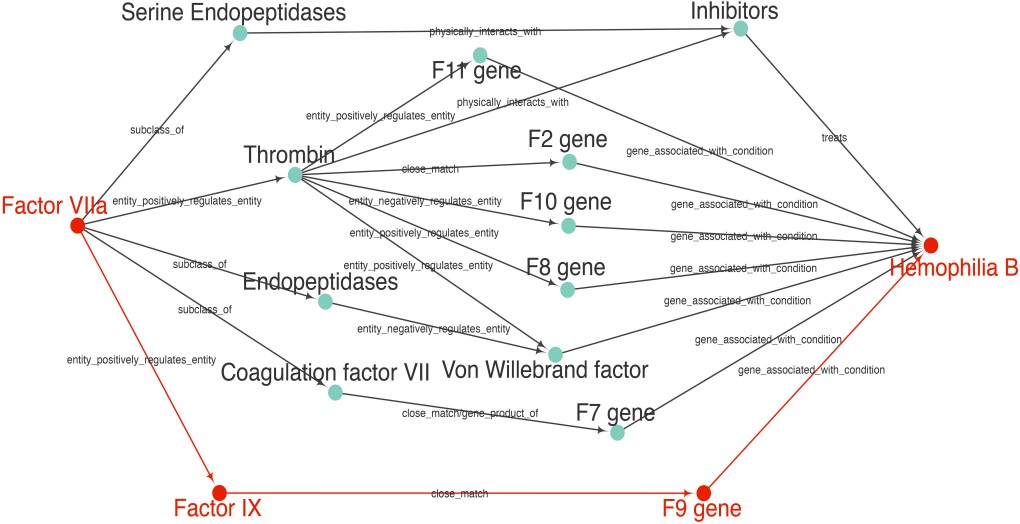

Figure 8: A subgraph with the top 10 predicted 3-hop paths for Factor VIIa – Hemophilia B pair. The path highlighted with red color is biologically reasonable for explaining the treatment mechanism of Factor VIIa for Hemophilia B.

### C.3.2 HUNTINGTON'S DISEASE

Huntington's disease (HD) is a rare neurogenetic disorder that typically occurs in midlife with symptoms of depression, uncontrolled movements, and loss of cognitive ability. The drugs for HD are mainly used for the treatment of its symptoms in abnormal movements (e.g., chorea) and psychiatric phenotype. We show ten highest-probability drugs/treatments predicted by the proposed drug repur-

Table 8: Top 10 predicted drugs/treatments for hemophilia B (note that the drugs highlighted with red color are used in the training set while the remaining drugs are the top-rank, non-chemotherapeutic drugs without showing up in the training set).

| Rank | Drug/Treatment | Prob. | Publications |
|------|----------------|-------|--------------|
| 1 | Eptacog Alfa (rFVIIa) | 0.833 | (Croom & McCormack, 2008; Minno, 2015) |
| 2 | Nonacog Alfa (rFIX) | 0.803 | (Rendo et al., 2015) |
| 3 | Viral Vector | 0.780 | (Driessche et al., 2001) |
| 4 | Factor VIIa | 0.748 | (Croom & McCormack, 2008; Castaman, 2017) |
| 5 | Recombinant FVIIa (rFVIIa) | 0.724 | (Croom & McCormack, 2008; Castaman, 2017) |
| 6 | Thrombin | 0.709 | (Negrier et al., 2019) |
| 7 | Factor IX | 0.708 | (Goodeve, 2015) |
| 8 | Epicriptine | 0.702 | |
| 9 | Hyperbaric Oxygen | 0.660 | |
| 10 | Triamcinolone | 0.649 | |

Table 9: Top 5 predicted drugs/treatments used in the training set (highlighted with red color) and the top 5 non-chemotherapeutic predicted drugs/treatments that are not in the training set for Huntington's Disease.

| Rank | Drug/Treatment | Prob. | Publications |
|------|----------------|-------|--------------|
| 1 | Pimozide | 0.939 | (Arena et al., 1980; Videnovic, 2013) |
| 2 | Therapeutic Agent | 0.939 | |
| 3 | Olanzapine | 0.938 | (Paleacu et al., 2002; Squitieri et al., 2001) |
| 4 | Riluzole | 0.935 | (Group, 2003) |
| 5 | Antipsychotic Agent | 0.932 | (Unti et al., 2017) |
| 10 | Risperidone | 0.893 | (Coppen & Roos, 2017; Duff et al., 2008) |
| 14 | Entinostat | 0.888 | (Shukla & Tekwani, 2020) |
| 15 | Primaquine | 0.887 | |
| 17 | Isradipine | 0.884 | (Miranda et al., 2019) |
| 19 | Amifampridine | 0.882 | |

posing model (Sec. 4.1) in Table 9. Like the case study of Hemophilia B, we exclude the chemotherapeutic drugs in the predicted drug candidate list. Since the top 8 predicted drugs/treatments in the predicted drug list are all in the training set, we only display the top 5 of them. The rest five in Table 9 are those with the highest probabilities in the predicted drug list after excluding the drugs used in the training set and all chemotherapeutic drugs. From Table 9, We can see that many top-rank predicted drugs are supported by some publications for the potential treatment of HD's symptoms. In order to evaluate the predicted paths for the predicted drugs/treatments that are not in the training set, we show the subgraphs with the top 10 predicted paths for each of the top 5 non-chemotherapeutic predicted drugs/treatments that are not in the training set in Figure 9. From these predicted paths, we can see that most of them are biologically relevant. For example, the subfigure (a) of Figure 9 shows that Risperidone is predicted to be useful for HD by decreasing the activity of the genes associated with the 5-Hydroxytryptamine receptor (e.g., HTR1A, HTR2A, HTR2C, HTR7) and dopamine receptor (e.g., DRD1, DRD2, DRD3) which have been proven to be involved in the pathogenesis of depressive disorders (Yohn et al., 2017; Delva & Stanwood, 2021) and depressive symptom is one of the important characteristics of HD (Coppen & Roos, 2017). Entinostat is predicted to inhibit the functions of histone deacetylase genes (e.g., HDAC1, HDAC2, HDAC6) to treat HD (see subfigure (b) of Figure 9). One of the predicted 3-hop MOA paths ("Entinostat" → "decreases activity of" → "HDAC1 gene" → "interacts with" → "Histone H4" → "gene associated with condition" → "Huntington's disease") are supported by the previous research (Shukla & Tekwani, 2020; Yu et al., 2009). Primaquine is predicted to act on the NQO2 gene and the IKBKG gene to play a therapeutic role in neurodegenerative disease which is reported in (Voronin et al., 2021; Singh & Singh, 2020). Isradipine is predicted to have a potential therapeutic effect for HD by mainly regulating the genes of the Calcium Voltage-Gated Channel (e.g., CACNA1S, CACNA1D, CACNA1C, CACNB2, CACNA2D2) which might be associated with the symptoms (e.g., chorea, depression and dementia) of HD (Yagami et al., 2012) while Amifampridine is predicted to regulate the genes of the Potassium Voltage-Gated Channel to associate with HD (Noh et al., 2019). All these examples indicate that the predicted KG-based MOA paths can explain the mechanism of repurposed drugs to some extent.

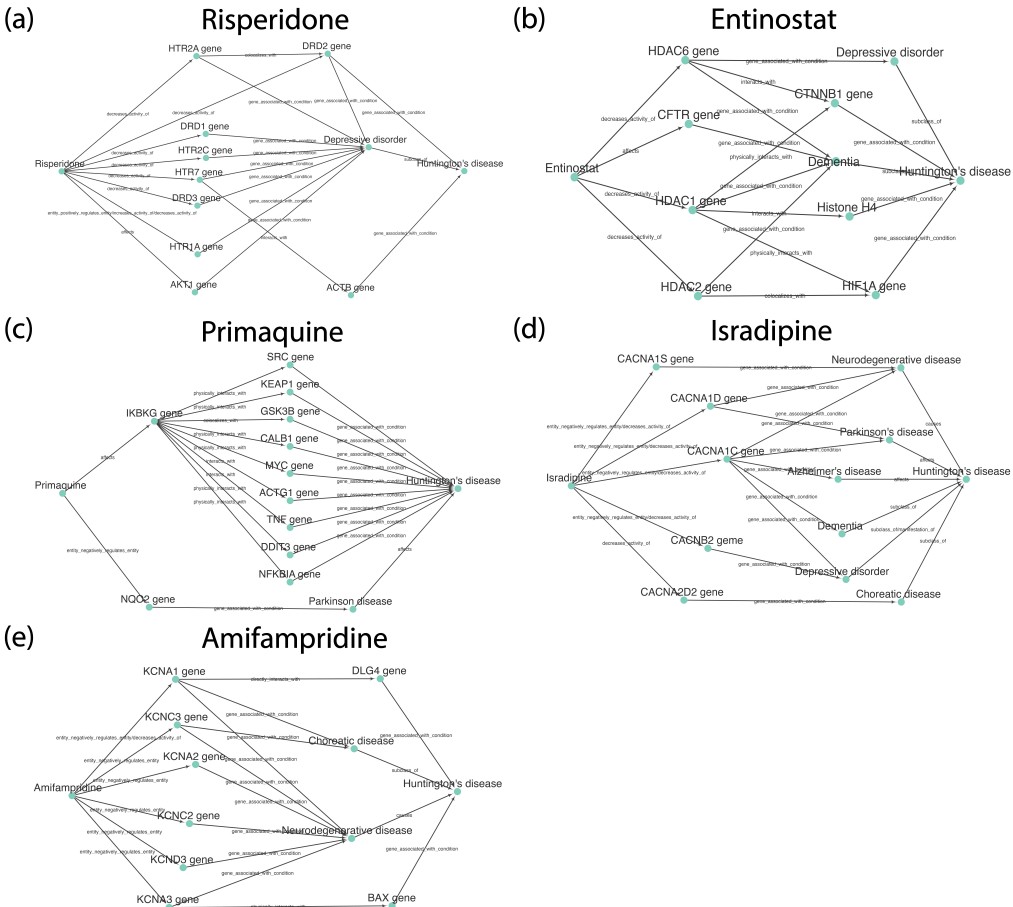

Figure 9: Top 10 predicted 3-hop paths (integrated into subgraphs) for top 5 non-chemotherapeutic predicted drugs/treatments that are not used in the training set for Huntington's disease.

