# OpenReview forum: "Predicting Drug Repurposing Candidates and Their Mechanisms from A Biomedical Knowledge Graph"
_ICLR.cc/2023/Conference — Submitted to ICLR 2023_

### Official Review · Reviewer_i213 · 2022-10-19

**Confidence:** 3
**Correctness:** 3
**Technical Novelty And Significance:** 3
**Empirical Novelty And Significance:** 2
**Recommendation:** 6

**Clarity, Quality, Novelty And Reproducibility:**

$Clarity\ and\ Quality$:

Ths paper is well written. The proposed computational drug repurposing approach is well clarified. Important baselines are missing, and some performance advantages are not explained.

$Novelty$:

The novelty is limited. As far as I'm concerned, the authors neither propose a new problem (idea) nor a new techique (technical).

$Reproducibility$:

The proposed method is reproducible based on textual descriptions.

**Strength And Weaknesses:**

$Strengths$:

1. The proposed approach is sound and well clarified. The whole manuscript is friendly to readers who are not familiar with drug repurposing. Enough pre-processing and implementation details are provided for enhancing the reproducibility. The visualization results of explored paths are appreciated.

2. The performance improvements are obvious when using MRR and Hit@K as evaluation metrics, showing that the number of false positive predictions is greatly reduces.

$Weaknesses$:

1. The novelty is limited. Knowledge graph (KG) based drug repurposing is not a new problem (see the second paragraph in sec. Related Work).  The use of graph neural networks (GraphSAGE in this paper) and Random Forest for treatment probability prediction is also not novel. The biomedical explanation exploration methodology, i.e., ADversarial Actor-Critic (ADAC) reinforcement learning with demonstration paths, was initially proposed by (Zhao et al., 2020) for path exploration. I agree that how to appropriately integrate existing machine learning techniques is an important issue for interdisciplinary research. Thus, the authors should clarify the novelty of ideas or techniques in the rebuttal.

2. The involved baselines are too simple and not optimized for the drug repurposing problem. The authors have mentioned that there exist a number of computational approached designed for drug repurposing in sec. Introduction and sec. Related Work. However, none of these methods are included for comparison. Since KG-based drug repurposing is not a new problem, the authors should include some recent methodologies as baselines, which could better demonstrate the strengths of proposed method.

3. Experimental results in Table 1 are not explained. In fact, I am surprised that a simple combination of GraphSAGE and Random Forest can outperform those baselines by such large margins (nearly 10% on MRR and Hit@3, and over 10% on Hit@5). The authors should at least give some explanations about where these improvements come from.

Minor points:

1. When the authors listed three groups of existing drug repurposing methods (in the second paragraph of sec. Introduction), some references should be added to help readers get an overview of the current research.

2. The name mechanism of action (MOA) is kind of confusing. If this is a previously used phrase, please add a citation. If not, the authors should replacing it with another more intuitive name.

3. The related work section should include some discussions about the differences between the proposed framework and previous related methods.

4. In equation 1, the authors claimed "$v$ is a node that co-occurs with $u$ in fixed-length random walks." But how to choose $z_v$ is not clarified.

**Summary Of The Paper:**

This paper addresses an important issue in drug repurposing methods, which is how to find biologically reasonable paths between drugs and potentially targeted diseases. The authors present a computational framework that can not only predict the treatment probabilities between drugs and diseases but also produce path-based biomedical explanations. To perform the first task (i.e., treatment probability prediction), the proposed framework uses GraphSAGE to integrate neighborhood information and apply Random Forest to predict treatment probabilities on top of the outputs of GraphSAGE. For the second task, an adversarial actor-critic reinforcement learning approach with demonstration paths are used to mine explainable paths between drugs and diseases. The experimental results are promising, indicating the proposed method is able to discover possible drug-disease pairs with much lower false positives.


**Summary Of The Review:**

I lean upon a borderline reject due to the limited novelty and a lack of domain-related baselines. However, if the authors could clarify these problems in the rebuttal, I would be happy to raise my score.

---

> ### Author Response · Authors · 2022-11-11
> **Response to reviewer i213 (Part 1)**
>
> # weakness 1
>
> Thanks for your concern about the novelty of our approach. Although the KG-based drug repurposing prediction is not a novel problem, the path-based explanation for these drug repurposing is still an unsolved issue by current methods, especially with regard to scalability to a  massive biomedical knowledge graph. To our best knowledge, our work is the first research for this specific issue.
>
> Although our approach seems like a combination of several existing models, applying these methods to this problem is far from trivial. For example, how to define biomedically meaningful demonstration paths to guide the search of RL model? How to provide meaningful intermediate reward to amplify the model training? How to utilize one of the largest biomedical knowledge graphs (over 6M nodes with the permissive and detailed definitions of drugs and disease such as “hypertension, mild”, “hypertension, severe”, “hypertension, renal” etc.) in drug repurposing research, which involves the difficulties in the identification of the same biological concepts with various names from knowledge providers in large scale and graph pre-processing in a biomedically meaningful and engineering applicable way.
> In this paper, we provide a decent answer to the above questions. We propose new ideas to amplify the ability of the RL model in biologically meaningful path searching by utilizing the biologically meaningful demonstration paths and pre-trained drug-repurposing model probability as terminal reward. We incorporate these ideas into the appropriate models (e.g. GraphSAGE, Random Forest and ADAC models) and then make them applicable to the explainable drug repurposing problem at large data scale and complexity, which are the main novelty of this paper.
>
> Our advancements to these models include:
> - In Zhao et al 2020, the authors utilize the shortest paths or the random-walk-based paths as demonstration paths for model training. These strategies do not make sense to find the correct paths for drug repurposing prediction explanation, especially in a massive knowledge graph. We are innovative in using a knowledge-based and publication-based (e.g., the known drug-target interactions and the PubMed publication based Normalized Google Distance) method to extract demonstration paths for drug repurposing explanation purpose. We demonstrate the efficacy of our proposed demonstration extraction method in Table 2.
>
> - The initial ADAC model proposed by Zhao et al 2020 uses an indicator function as the terminal reward function, i.e. 1 for the correct diseases and 0 otherwise. However, this might result in high false negatives (some potential diseases not in the training set will also be considered as true negatives) and misdirect the agent in finding the potential paths for drug repurposing, especially in a massive knowledge graph which might contain more than hundreds of thousands of paths between given drug-disease pairs. We address this issue by constructing a drug-repurposing model that predicts the potential probability of a given drug in treating a given disease and then use it as a reward function in the RL model. Furthermore, to increase the prediction accuracy of the proposed drug repurposing model, we leverage both the node attribute information and graph topology information by using the node attribute embeddings as the initial node feature for GraphSAGE model. We also demonstrate that this modification can efficiently improve the model performance in Table 7.
>
> # weakness 2
> Thanks for your suggestions. Actually, we have already used almost all the common KG-based models as baselines. It is true that there are many computational approaches such as those leveraging chemical structure, drug perturbations of gene expression, phenotype screening for drug repurposing purpose. As we mentioned in sec. Introduction, they are less cost-efficient than the KG-based approaches because they have to incorporate the experimental analysis like the chemical analysis or sequence data analysis. The most common KG-based models for drug repurposing are the common graph-based models, such as GCN, TransE, DistMult, CompIEx, ConvE used in Ioannidis et al. 2020, Zhang et al. 2021,  Zhang and Che, 2021.
>
> Also, as we described in Section 5.4,  for those existing baselines that also utilize the KG paths for drug repurposing purpose, they either require external knowledge (e.g., the confidence scores of relation edges) or cannot be trained within a reasonable time (e.g. within two weeks) due to the scale and complexity of the knowledge graph.

---

> > ### Comment · Reviewer_i213 · 2022-11-11
> > **Reply to Paper1980 Authors**
> >
> > Thanks for your clarification. Some of my concerns have been resolved. However, I still have following questions.
> >
> > 1.  Have the rest of baselines (besides 2-class GraphSAGE+RF) used 2-class labels for training (i.e., {treat, not treat})? If the answer is yes, I think the improvements mainly come from the use of RF (compared to GAT) and the introduction of the third class (i.e., unknown).
> >
> > 2. What are the physical meanings of MRR, Hit@1, Hit@3, and Hit@5? For instance, what is the benefit of achieving a high MRR score?
> >
> > 3. From my perspective, I think the "demonstration path" can be regarded as the "ground truth" path. Thus, it can provide large performance improvements, compared to ADAC RL (as shown in Table 2). Since the ADAC part is your main technical contribution, I think thorough ablation studies are needed to demonstrate the superiority of your modifications to the naive ADAC. For example, how to define biomedically meaningful demonstration paths to guide the search of RL model? How to provide meaningful intermediate reward to amplify the model training?
> >
> > 4. How do you split the train/val/test sets? A random split, I guess?
> >
> > Overall, I think this is a borderline paper. I appreciate the authors' great efforts. However, the technical novelty is somewhat limited and has not been well clarified. Currently, I lean upon a weak accept but I think the draft requires major revision to make the contents more clarified.

---

> > > ### Author Response · Authors · 2022-11-12
> > > **Response to the further questions of reviewer i213**
> > >
> > > Thanks for your further questions. Please see our responses to them below:
> > >
> > > > 1. Have the rest of baselines (besides 2-class GraphSAGE+RF) used 2-class labels for training (i.e., {treat, not treat})? If the answer is yes, I think the improvements mainly come from the use of RF (compared to GAT) and the introduction of the third class (i.e., unknown).
> > >
> > > Yes, the rest of baselines (besides 2-class GraphSAGE+RF) used 2-class labels for training. We illustrate this in the paragraph “Drug repurposing prediction” in Sec. 5.2. Based on the comparison in both Table 1 and Table 7, we conclude that the overall improvement (e.g., accuracy-based metrics  and ranking-based metrics) of our model mainly comes from three modifications: incorporating node attribute embeddings as initial features for GraphSAGE (concluded from Table 7); the combination of GraphSAGE and RF (compared to GAT and GraphSAGE+link); the usage of negative sampling (the introduction of “unknown” class)
> > >
> > > > 2. What are the physical meanings of MRR, Hit@1, Hit@3, and Hit@5? For instance, what is the benefit of achieving a high MRR score?
> > >
> > > In short, achieving a higher MRR, Hit@1, Hit@3, and Hit@5 means that the model gives higher ranks to the “correct”  (according to the training/validation/test data) diseases and avoids false positives. If we sort the candidate diseases based on the probability of the “treat” class predicted by the model in descending order, a higher MRR means the “correct” diseases are higher on the list. A high Hit@1 means a correct disease was frequently assigned the highest probability amongst all diseases for that given drug. A high Hit@3 means a correct disease was often amongst the top 3 highest probability diseases. A high Hit@5 means a correct disease was often amongst the top 5 highest probability diseases. They illustrate the model’s potential in identifying correct indications of drugs out of all the candidate diseases.
> > >
> > > > 3. From my perspective, I think the "demonstration path" can be regarded as the "ground truth" path. Thus, it can provide large performance improvements, compared to ADAC RL (as shown in Table 2). Since the ADAC part is your main technical contribution, I think thorough ablation studies are needed to demonstrate the superiority of your modifications to the naive ADAC. For example, how to define biomedically meaningful demonstration paths to guide the search of RL model? How to provide meaningful intermediate reward to amplify the model training?
> > >
> > > Thanks for your suggestion. We did compare our ADAC part to an ADAC model without using our “demonstration path” (please see “ADAC RL w/o DP” in Table 2). As shown in the table, using the “demonstration path” leads to a big performance improvement (e.g., MPR 72.96 -> 94.59, MRR 0.015 -> 0.109). We also clearly define how to generate biomedically meaningful demonstration paths in Appx. A2 (Due to the page limit, we’re sorry that we cannot put this description in the main text) and how to incorporate them into the ADAC model and amplify the model training in the paragraphs “Path discriminator network” and “Meta-Path discriminator network” in Sec. 4.2.2.
> > >
> > > > 4. How do you split the train/val/test sets? A random split, I guess?
> > >
> > > The split ratio 8/1/1 is random, but for each drug individually. For example, let’s say drugA has 10 known diseases that it treats, (e.g., drugA-disease1, …, drugA-disease10), 8 pairs are randomly split into the training set, 1 pair is to the validation set, 1 pair to the test set. This is done so that the model can be exposed to every drug in the training set, since our goal is to identify new indications from known, existing mechanisms of action.

---

> > > > ### Comment · Reviewer_i213 · 2022-11-18
> > > > **Reply to Paper1980 Authors**
> > > >
> > > > Dear authors,
> > > >
> > > > Thanks for your clarification. I have raised my score to 6 since your response has addressed most of my concerns.
> > > >
> > > > Good luck!
> > > >
> > > > Best,

---

> ### Author Response · Authors · 2022-11-11
> **Response to reviewer i213 (Part 2)**
>
> # weakness 3
>
> Sorry for lack of sufficient explanation due to the page limit. According to the comparison with the 2-class version of GraphSAGE with Random Forest shown in Table 1 and the ablation experiment shown in Table 7, the significant improvement mainly comes from the usage of negative sampling (e.g., we trained a trinary classifier that tries to classify the drug-disease pairs into three classes "treat", "not treat" and "unknown". By this way, we encourage the model to assign higher probabilities to the observed true positive drug-disease pairs.) and the incorporation of node attribute embedding as the initial feature in the GraphSAGE model (mentioned in the second paragraph of sec. 4.1).
>
> ## minor point 1
>
> The three-group drug repurposing classification method is based on the paper Dhir et al. 2020.  We cite this paper in our paper and hence we didn’t add the references to each of those drug repurposing methods due to the page limit.
>
> ## minor point 2
>
> Sorry for the confusion of the term “mechanism of action (MOA)”.  This term is used frequently in the domains of biomedicine and pharmacology, referring to the specific biochemical interaction through which a drug produces a pharmacological effect. We have already added a citation (Davis, 2020) in the updated version of the paper for this term.
>
> ## minor point 3
>
> Sorry for lack of sufficient discussion about the differences between the proposed framework and previous methods. Due to the page limit, we can only mention the features of the previous related methods and their deficiency that we want to address. We did a simple discussion in the “baselines” paragraphs in sec. 5.3 and 5.4. We will add more in the camera-ready version if the manuscript is accepted.
>
> ## minor point 4
>
> Sorry for missing this clarification. For each node $v$ in the knowledge graph, we used the 100 random walk length and 10 numbers of walk each starting from the node $v$ to choose zv in the knowledge graph. We have already added this description in the Appx. B.1 in the updated version of the paper.

---

### Official Review · Reviewer_4uZj · 2022-10-23

**Confidence:** 3
**Correctness:** 3
**Technical Novelty And Significance:** 2
**Empirical Novelty And Significance:** 2
**Recommendation:** 5

**Clarity, Quality, Novelty And Reproducibility:**

*Clarity*

The paper is well-written and the ideas are clearly explained

*Novelty*

All techniques described in the paper already exist. The main contribution relies on the combination of the methods and the formulation of ADAC model for drug repurposing.

*Reproducibility*

The authors provided their code for reproducing the experiments and also described the analyses in details.

**Strength And Weaknesses:**

The approach combines existing methods: PubMedBERT for defining the node attributes in the knowledge graph, GraphSAGE for learning node embeddings, random forest for predicting drug efficacy, and ADversarial Actor-Critic (ADAC) Reinforcement learning model for predicting drug mechanism of action paths.



The novelty of the work relies on the combination of these methods. In addition, it formulates the ADAC model properly for the specific problem.



The proposed approach makes sense, and seems to work well for both problems addressed by the paper: prediction of drug efficacy and prediction of drug mechanism-of-action paths.



It is important to notice that some of the competitors rely on related ideas (reinforcement learning for predicting drug mechanism of action pathways), but they use different models.



The manuscript is well written, and the authors provided the code used for their approach.



I reviewed this paper for NeurIPS 2022. The authors already answered many of my original concerns and improved the paper. However, I still have a concern on the evaluation and some minor comments that I detail below.



1) Evaluation

3.1. In Section 5.2 and Figure 4, the authors say that they evaluated the methods in a dataset composed of the validation and test sets. Could the authors provide the evaluation only on the test set?



2) Minor comments

- The authors refer to a bioRxiv version of the RTX-KG2 paper. I saw that the paper was recently published on BMC Bioinformatics:

Wood, E.C., Glen, A.K., Kvarfordt, L.G. et al. RTX-KG2: a system for building a semantically standardized knowledge graph for translational biomedicine. BMC Bioinformatics 23, 400 (2022). https://doi.org/10.1186/s12859-022-04932-3

 Could the authors update the reference?

- It looks like references are not ordered by the last names of the authors. It was difficult to look up for the references.

**Summary Of The Paper:**

The authors proposed an approach for drug repurposing based on a biomedical knowledge graph. It not only predicts drug efficacy against a disease, but it also finds paths that could provide a biological explanation for the predictions. They compared their method against competitors and showed their performance in predicting drugs and in recapitulating curated drug mechanism

**Summary Of The Review:**

The paper combines existing approaches on a non-trivial way for solving an interesting and difficult problem.

For accepting the paper, I think the evaluation should be improved.

---

> ### Author Response · Authors · 2022-11-11
> **Response to Reviewer 4uZj (part 1)**
>
> Thanks for your comment. We rerun the evaluation step only on the test set and show the results below:
>
> ## Table 1 updated with only the test set
> | **Model**                   | **Accuracy**   | **Macro F1 score** | **MRR** | **Hit@1** | **Hit@3** | **Hit@5** |
> |---------------------------|:--------------:|:------------------:|:-------:|:---------:|:---------:|:---------:|
> | TransE                      | 0.707          | 0.708              | 0.280   | 0.117     | 0.301     | 0.449     |
> | TransR                      | 0.857          | 0.854              | 0.301   | 0.123     | 0.343     | 0.518     |
> | RotatE                      | 0.704          | 0.704              | 0.252   | 0.075     | 0.283     | 0.448     |
> | DistMult                    | 0.555          | 0.495              | 0.173   | 0.041     | 0.142     | 0.259     |
> | ComplEx                     | 0.624          | 0.460              | 0.131   | 0.020     | 0.105     | 0.193     |
> | ANALOGY                     | 0.594          | 0.465              | 0.180   | 0.045     | 0.145     | 0.272     |
> | SimplE                      | 0.599          | 0.472              | 0.163   | 0.038     | 0.137     | 0.241     |
> | GAT                         | **0.936**          | **0.934**              | 0.003   | 0.001     | 0.001     | 0.001     |
> | GraphSAGE-link              | 0.919          | 0.915              | 0.002   | 0         | 0         | 0         |
> | GraphSAGE+logistic          | 0.791          | 0.784              | 0.002   | 0         | 0         | 0         |
> | GraphSAGE+SVM               | 0.807          | 0.793              | 0.002   | 0         | 0         | 0         |
> | 2-class GraphSAGE+RF        | 0.929          | 0.925              | 0.271   | 0.177     | 0.310     | 0.381     |
> | 3-class GraphSAGE+RF (ours) | 0.935 (0.930*) | 0.923 (0.926*)     | **0.356**   | **0.206**     | **0.407**     | **0.522**     |
>
> ## Table 2 updated with only the test set
> | **Model**            | **MPR** | **MRR** | **Hit@1** | **Hit@10** | **Hit@50** | **Hit@100** | **Hit@500** |
> |----------------------|---------|---------|-----------|------------|------------|-------------|-------------|
> | MultiHop             | 61.400% | 0.027   | 0.017     | 0.042      | 0.067      | 0.118       | 0.345       |
> | ADAC RL w/o DP       | 72.965% | 0.015  | 0.008     | 0.017      | 0.067      | 0.160       | 0.403       |
> | ADAC RL w/ DP (ours) | **94.696%** | **0.109**   | **0.059**     | **0.193**      | **0.496**      | **0.613**       | **0.849**       |
>
> ## Table 7 updated with only the test set
> | **Model**                                         | **Accuracy** | **Macro F1 score** | **MRR** | **Hit@1** | **Hit@3** | **Hit@5** |
> |---------------------------------------------------|--------------|--------------------|---------|-----------|-----------|-----------|
> | 3-class GraphSAGE+RF w/o node attribute embedding | 0.909        | 0.891              | 0.150   | 0.029     | 0.138     | 0.243     |
> | 3-class GraphSAGE+RF w/ node attribute embedding  | **0.935**        | **0.923**              | **0.356**   | **0.206**     | **0.407**     | **0.522**     |

---

> ### Author Response · Authors · 2022-11-11
> **Response to Reviewer 4uZj (part 2)**
>
> Since we can’t paste the figure, we convert the figure 4 result with only the test set to the table below (Note that three different “all nodes” replacement results are separated by “/”. The three values in each cell correspond to “Combined Replacement”, “Drug-rank-based Replacement”, and “Disease-rank-based Replacement”described in Appx. C2):
>
> ## Figure 4 result with only test set
> | **Model**                   | **MRR**           | **Hit@1**         | **Hit@3**         | **Hit@5**         | **Hit@10**         | **Hit@50**        |
> |:---------------------------:|:-----------------:|:-----------------:|:-----------------:|:-----------------:|:------------------:|:-----------------:|
> | TransE                      | 0.003/0.035/0.023 | 0/0.013/0.007     | 0.001/0.028/0.018 | 0.001/0.044/0.028 | 0.002/0.074/0.052  | 0.023/0.206/0.145 |
> | TransR                      | 0.004/0.01/0.012  | 0/0.001/0.002     | 0.001/0.006/0.006 | 0.002/0.009/0.011 | 0.004/0.017/0.020  | 0.026/0.095/0.097 |
> | RotatE                      | 0.002/0.021/0.015 | 0/0.005/0.003     | 0/0.018/0.008     | 0.001/0.024/0.016 | 0.003/0.039/0.034  | 0.011/0.135/0.095 |
> | DistMult                    | 0.001/0.006/0.008 | 0/0.001/0         | 0/0.003/0.005     | 0/0.004/0.010     | 0/0.009/0.018      | 0.001/0.042/0.064 |
> | ComplEx                     | 0.001/0.006/0.002 | 0/0/0             | 0/0.002/0.001     | 0/0.004/0.001     | 0/0.008/0.001      | 0/0.043/0.013     |
> | ANALOGY                     | 0.001/0.005/0.012 | 0/0.001/0.001     | 0/0.002/0.007     | 0/0.004/0.013     | 0/0.008/0.025      | 0.001/0.037/0.092 |
> | SimplE                      | 0.001/0.005/0.007 | 0/0.001/0         | 0/0.003/0.003     | 0/0.003/0.005     | 0/0.006/0.013      | 0/0.034/0.060     |
> | GraphSAGE-link              |     0.005/0.006/0.010          |          0.005/0.006/0.010         |       0.005/0.006/0.010            |      0.005/0.006/0.010             |          0.005/0.006/0.010          |         0.005/0.006/0.010          |
> | GraphSAGE+logistic          | 0/0/0             | 0/0/0             | 0/0/0             | 0/0/0             | 0/0/0              | 0/0/0             |
> | 2-class GraphSAGE+RF        | 0.013/0.101/0.053 | 0.005/0.066/0.028 | 0.010/0.108/0.053 | 0.014/0.129/0.067 | 0.024/0.164/0.099  | 0.081/0.284/0.199 |
> | 3-class GraphSAGE+RF (ours) | **0.022**/**0.135**/**0.078** | **0.010**/**0.092**/**0.044** | **0.018**/**0.141**/**0.079** | **0.023**/**0.171**/**0.105** | **0.040**/**0.212**/**0.144**  | **0.114**/**0.351**/**0.274** |
>
>
> Regarding the minor points:
>
> Could the authors update the reference for the database?
>
> - Thanks for your kind reminder. Since this paper was published in BMC Bioinformatics after we submitted our paper to ICLR 2023, we used the bioRxiv version. We have already updated it in the updated version of the paper. Thank you!
>
> It looks like references are not ordered by the last names of the authors. It was difficult to look up the references.
> Response:
>
> - We have already updated the reference list by the last names of the authors. Thanks!

---

### Official Review · Reviewer_cTJB · 2022-10-25

**Confidence:** 5
**Clarity, Quality, Novelty And Reproducibility:** 1. There are some minor grammar error…
**Correctness:** 2
**Technical Novelty And Significance:** 2
**Empirical Novelty And Significance:** 2
**Recommendation:** 3

**Details Of Ethics Concerns:**

The case studies contain some predicted drugs and their MOAs for Hem B or HD. If published, some detailed, rigorous reviews from bioinformatics experts are needed.

**Strength And Weaknesses:**

Strength
The paper uses existing ML methods for a real-world AI for Science problem.

Weaknesses
1. The paper is neither novel from ML (only uses existing, well-known methods) nor from biomedicine (the formulation of drug repurposing as link prediction is not rigorous and convincing; the biomedical knowledge graph comes from other bioinformatics papers).

2. While the paper has been claimed as a drug repurposing study, it lacks some necessary background information for this field.
   a. There are many AI for drug repurposing methods, such as chemical structure, drug-target interactions, drug perturbations of gene expression, cell and animal phenotypes, and clinical information. But the related work ignores all of them.
   b. Drug repurposing is one of the hardest problems in biomedicine, but it seems that (from Table 1) the paper reports an extremely high accuracy (0.934) for this task. It is necessary to compare the proposed method with the existing drug repurposing methods in the field, chemical structure, drug-target interactions, gene expression etc., and fully evaluate the method.

3. While the paper all depends on a "massive" biomedical knowledge graph, it doesn't provide enough information for evaluating its quality as the training/testing data.
   a. How to generate "true negative" data in the knowledge graph? In other words, how to prove the drug is NOT AND NEVER possible to treat a disease? In this sense, only RepoDB data is possible (negative is defined as failure or withdrawal in RCT). For each dataset, the paper needs to discuss how to get the true negative without any RCT (it seems a mission impossible). And if there exists such a pos/neg, any binary classification methods will do the job.
   b. The paper doesn't provide raw data for the review (those ids are hard to interpret). But from Table 8 and Table 9, it seems that the quality of the biomedical knowledge graph is poor. e.g., "therapeutic agent" and "antipsychotic agent" as a drug? The authors will need significant efforts to curate the knowledge graph.
   c. To avoid GIGO, the authors need to provide the full knowledge graph with all detailed information and reliable references.

4. There are some major problems in the experimental design.
   a. The paper uses only cross-validation (8:1:1) for a link prediction problem, which leaks information. Since RepoDB is the ONLY reliable TP/TN resource, I suggest the authors to use the other 3 datasets to predict links in RepoDB - it avoids dataset information leak.
   b. Also, I suggest the authors to leave-drug-out (instead of leave-link-out) when reporting all the scores - as the abstract mentions "identify new indications... of drugs/compounds". The paper needs to report the performance of predicting indications for each drug.
   c. Moreover, it would be great to leave-drug-class-out or leave-similar-drug-out to rule out the me-too drugs (I will not be surprised if pravastatin, atorvastatin, fluvastatin or oxprenolol, penbutolol, pindolol treat the same group of diseases) in the evaluations. Only doing this will demonstrate the true contribution of this paper.
   d. What does the "probability" mean in the table 8 and table 9? For example, 0.882 for amifampridine (a muscle drug): does it mean if a drug company launches amifampridine in RCT, there is 88.2% chance to be successful or if amifampridine be used for 1000 Huntington patients there are 882 patients will be saved? Some discussions and justifications are needed.

5. (minor) The references are not sorted based on ICLR template.

**Summary Of The Paper:**

The paper proposes a computational drug repurposing framework that not only predicts the "treatment probabilities" between drugs and diseases but also predicts the path-based, "testable" mechanisms of action as their biomedical explanations, all based on a "massive" biomedical knowledge graph. Specifically, the paper uses an existing GraphSAGE model (NIPS 2017) with RF for the predictions and uses an adversarial actor-critic RL (SIGIR 2020) to predict MOA paths. The paper uses existing methods for an AI for Science problem.

**Summary Of The Review:**

The paper uses some existing, well-known methods for a biomedical, high-profile drug repurposing problem. The paper is neither novel from ML nor from biomedicine. There are also some major concerns in the data and experiments.

---

> ### Author Response · Authors · 2022-11-11
> **Response to Reviewer cTJB (part 1)**
>
> # weakness 1
>
> Thanks for your comment.
>
> We think it may be unfair to describe our method as “use existing methods for an AI for Science problem.”
>
> From an ML perspective, although the proposed framework is a combination of several existing methods (GraphSAGE, Random Forest, and the ADAC model), applying these methods to this problem is far from trivial. For example, how to define biomedically meaningful demonstration paths to guide the search of RL model? How to provide meaningful intermediate reward to amplify the model training? How to utilize one of the largest biomedical knowledge graphs (over 6M nodes with the permissive and detailed definitions of drugs and disease such as “hypertension, mild”, “hypertension, severe”, “hypertension, renal” etc.) in drug repurposing research, which involves the difficulties in the identification of the same biological concepts with various names from knowledge providers in large scale and graph pre-processing in a biomedically meaningful and engineering applicable way.
>
> In this paper, we provide a decent answer to the above questions. We propose new ideas to amplify the ability of the RL model in biologically meaningful path searching by utilizing the biologically meaningful demonstration paths and pre-trained drug-repurposing model probability as terminal reward. We incorporate these ideas into the appropriate models (e.g. GraphSAGE, Random Forest and ADAC models) and then make them applicable to the explainable drug repurposing problem at large data scale and complexity, which are the main novelty of this paper.
>
> From a technical perspective, our advancements include:
>
> - In Zhao et al 2020, the authors utilize the shortest paths or the random-walk-based paths as demonstration paths for model training. These strategies do not make sense to find the correct paths for drug repurposing prediction explanation, especially in a massive knowledge graph. We are innovative in using a knowledge-based and publication-based (e.g., the known drug-target interactions and the PubMed publication based Normalized Google Distance) method to extract demonstration paths for drug repurposing explanation purpose. We demonstrate the efficacy of our proposed demonstration extraction method in Table 2.
>
> - The initial ADAC model proposed by Zhao et al 2020 uses an indicator function as the terminal reward function, i.e. 1 for the correct diseases and 0 otherwise. However, this might result in high false negatives (some potential diseases not in the training set will also be considered as true negatives) and misdirect the agent in finding the potential paths for drug repurposing, especially in a massive knowledge graph which might contain more than hundreds of thousands of paths between given drug-disease pairs. We address this issue by constructing a drug-repurposing model that predicts the potential probability of a given drug in treating a given disease and then use it as a reward function in the RL model. Furthermore, to increase the prediction accuracy of the proposed drug repurposing model, we leverage both the node attribute information and graph topology information by using the node attribute embeddings as the initial node feature for GraphSAGE model. We also demonstrate that this modification can efficiently improve the model performance in Table 7.
> From biomedicine perspective, we are the first to do drug repurposing based on a massive knowledge graph RTX-KG2, which, to our best knowledge, is one of the most comprehensive and open-source biomedical knowledge graphs and has been commonly used in the Biomedical Data Translator Project (https://ncats.nih.gov/translator/projects) funded by the NIH National Center for Advancing Translational Science (NCATS) and has been widely used in its related research. Therefore, unlike the previous drug repurposing work, we think one of contributions is proposing a framework that is scalable to a large knowledge graph with more biomedical connections.

---

> ### Author Response · Authors · 2022-11-11
> **Response to Reviewer cTJB (part 2)**
>
> # weakness 2
> It is true that there are many AI methods for drug repurposing such as molecular docking based approach, GWAS-based approach, chemical structure-based approach. However, the strength of knowledge graph (KG)-based approaches is that it is comparably more cost-efficient as we mentioned in sec. Introduction because plenty of curated biomedical data can be accessed from different open-source databases such as DrugBank, ChEMBL, HMDB, and they are also updated quickly.
>
> In addition, we demonstrate in Table 1 that our proposed KG-based drug repurposing approach is better in comprehensively considering both accuracy and low false positives compared with other common KG-based approaches, especially in such a large knowledge graph. We do not aim to present the definitive approach to computational drug repurposing, but rather demonstrate how emerging knowledge graphs can be successfully leveraged for this task.
>
> In our study, the framework we propose focuses on drug repurposing predictions based on the biomedical knowledge graph and tries to further explain the drug mechanism via the KG-based mechanism of action paths. It should be fair to only compare our method against those KG-based drug repurposing approaches. Therefore, to our best knowledge, the baseline models that we compare in Table 1 are almost all common KG-based drug repurposing methods.These methods are quite commonly used in the recent publications that did similar work like us such as Ioannidis et al. 2020 [1], Zhang et al. 2021 [2],  Zhang and Che, 2021 [3].
>
> [1] Ioannidis, Vassilis N, et al. “Few-shot link prediction via graph neural networks for covid-19 drug-repurposing.” arXiv preprint (2020b).
>
> [2] Zhang, Rui, et al. “Drug repurposing for covid-19 via knowledge graph completion.” Journal of Biomedical Informatics (2021).
>
> [3] Zhang, Xiaolin and Chao Che. “Drug Repurposing for Parkinson’s Disease by Integrating Knowledge Graph Completion Model and Knowledge Fusion of Medical Literature.” Future Internet (2021).
>
>
> # weakness 3
> The “massive” biomedical knowledge graph RTX-KG2 has been peer reviewed recently and published in the journal BMC Bioinformatics (https://bmcbioinformatics.biomedcentral.com/articles/10.1186/s12859-022-04932-3). Also, this knowledge graph is one of the key knowledge providers used in the Biomedical Data Translator Project (https://ncats.nih.gov/translator/projects) funded by the NIH National Center for Advancing Translational Science (NCATS) and has been widely used in its related research. To our best knowledge, most of the data underlying this knowledge graph has been previously published and human curated. The aggregation itself is the novel aspect of this graph.
>
> In addition, the data we used for training and test sets, as we mentioned in Appx. A3.1, are collected from four curated data sources: MyChem, SemMedDB, NDF-RT and RepoDB. They are all human-curated or supported by previous literature. The “true negative” data used in our study is defined as those drug - disease pairs with “contraindications” or “no effect” relations collected from these four datasets (described in  Appx. A3.1). It is impossible to evaluate all these drugs one by one via numerous randomized controlled trials (RCTs) because it is quite expensive and such datasets to our knowledge don’t exist, even for RepoDB. In fact, the RepoDB data itself is also drawn from those curated databases such as DrugCentral and ChinicalTrais.gov. The MyChem data (https://docs.mychem.info/en/latest/doc/data.html#data-sources) and NDR-RT (https://bioportal.bioontology.org/ontologies/NDFRT) are indeed obtained from these curated databases or fully human curated. The SemMedDB data is one of the most comprehensive literature-derived sources that has been approved to be useful for drug repurposing (e.g, https://www.ncbi.nlm.nih.gov/pmc/articles/PMC7869625/). Therefore, we believe using these data for training and test sets are reliable.
>
> Regarding the name of IDs used in the datasets, we provide the name mapping files for each of the datasets for you to better understand their underlying science. We’ve updated the training data folder and put a new folder (e.g., “translated_to_names”) that contains the names for each datasets. The name "therapeutic agent" (ID: UMLS:C1611640) and "antipsychotic agent" (ID: CHEBI:35476) from Table 8 and Table 9 seem not like a drug name but they are indeed considered as drugs according to the Unified Medical Language System (UMLS) and CHEBI. Please refer to https://www.ebi.ac.uk/chebi/searchId.do?chebiId=CHEBI:35476 as an example. It is manually annotated by the ChEBI Team. We defer to the experts who curate UMLS and CHEBI to judge what qualifies as a “drug.”
>
> We did not append this knowledge graph to submission due to its size (~6M nodes and ~39M edges). All references can be found at: https://github.com/RTXteam/RTX-KG2#what-data-sources-are-used-in-kg2

---

> > ### Comment · Reviewer_cTJB · 2022-11-21
> > **How to generate "true negative" data in the knowledge graph?**
> >
> > The explanations of "true negative" are not sufficient. While it makes sense for the authors to use RepoDB to define "negative" drug-disease pairs, it is completely unclear how MyChem (26,795 true negatives), SemMedDB (11 true negatives), and NDF-RT (5,119 true negatives) get such "negative" edges. Again, "how to prove the drug is NOT AND NEVER possible to treat a disease" without RCTs? Since it is an AI for Science paper and ALL experiments DEPEND ON such label definitions, the authors will need to pay more attention to the raw data to avoid GIGO problems. Very detailed information on how each dataset defines positive and negative edges is needed.

---

> > > ### Author Response · Authors · 2022-11-22
> > > **Recall that "contraindicated" implies "does not treat"; RCT results are included**
> > >
> > > The training data _does_ include results from RCTs, as well as true negatives that are so clearly true negatives that an RCT is unnecessary. As detailed in the Appendix section A.3.1, MyChem and NDF-RT both give human-curated indications and contraindications and include human-parsed RCT results for both true positives and negatives. Additionally, we reasonably assume that "contraindicated for" logically implies "does not treat." An RCT is not possible in such cases as it is doubtful an IRB would approve of such a contraindicated treatment. As for SemMedDB, the "negatively treats" predicate _does indeed encompass randomized controlled trials_. Such studies publish their findings and report the results in their abstracts, so SemMedDB can be considered a _superset_ of RCT results (hence the publication bias towards true positives). Of course, this is a superset as other finding are also included (such as the aforementioned contraindications). We set rigorous thresholds in order to account for the possibility of noise introduced by the SemMedDB NLP process, hence increasing the quality of these SemMedDB true negatives and rely on the human curation (as is done in RCTs) for the remaining two sources (MyChem and NDF-RT).

---

> > > > ### Comment · Reviewer_cTJB · 2022-11-22
> > > > **It makes me more worried. Why not evaluate only on trusted, RCT edges?**
> > > >
> > > > Are there any examples for "true negatives that are so clearly true negatives that an RCT is unnecessary" and "in such cases as it is doubtful an IRB would approve of such a contraindicated treatment"? Please explain in detail how MyChem and NDF-RT generate "contraindicated for" labels for so many drug-disease pairs (26,795 in MyChem and 5,119 in NDF-RT). Does every negative pair have either RCT support (in this case, a failed or terminated RCT) or some literature specifically for explaining why "the drug is NOT AND NEVER possible to treat a disease"? How do human curates for those "contraindications"?
> > > >
> > > > If they are not as trusted as RCTs, the author will at least need to evaluate the algorithms based only on those RCT-based edges - as #4 suggested in my review. As an AI for Science paper, especially the data used is related to experimental results, any "reasonably assume"/"implies" in this field needs very rigorous sources and designs.

---

> > > > > ### Author Response · Authors · 2022-11-22
> > > > > **RCTs aren't run for clearly contraindicated drugs, and we don't present this work as clinically actionable, just a methodological advancement**
> > > > >
> > > > > Before replying to your direct questions, let us address your concern about “AI in Science”: indeed, if we had presented this tool as one to be used directly for clinicians treating patients, then we whole-heartedly agree that the selection of true positive/negative would need to be significantly more stringent, as would the post-prediction filtering in order to protect patient health. As we detail in our ethics statement in section 7, this is _not_ what we are proposing in this paper. We merely attempt to advance the field of ML/AI in biology by proposing an architecture that addresses a hard problem (describing drug mechanisms of action), is scalable for application to emerging very large data sets (big KGs), improves upon current approaches, provides encouraging results, and indicates fruitful directions for future ML/AI research.
> > > > >
> > > > >
> > > > > Indeed, we are not alone in utilizing human-curated and NLP-derived training data sets to train “AI in science” approaches. A small selection of such previous work specifically for drug repurposing includes:
> > > > > https://elifesciences.org/articles/26726
> > > > >
> > > > > https://www.sciencedirect.com/science/article/pii/S1532046421000253
> > > > >
> > > > > https://pubmed.ncbi.nlm.nih.gov/31797619/
> > > > >
> > > > > https://ieeexplore.ieee.org/document/8574025
> > > > >
> > > > > It would indeed be far more favorable to utilize _only_ RCT-derived training data. If you are aware of any repositories that contain studies which utilize contraindicated drugs for treatment of conditions, we would be grateful if you could point them out to us. In the meantime, we rely on peer-reviewed, published, open-source, information to train our models. These likely contains errors, misclassifications, and occasional lack of experimental support. However, any large-scale machine learning approach will need to be robust to such “noise” in the training data. Our results demonstrate that our approach, which may utilize sub-par training data in your estimation, still indicates promising directions for future ML/AI research in this challenging area.
> > > > >
> > > > >
> > > > > To your specific questions:
> > > > >
> > > > >
> > > > > 1.	Are there any examples for "true negatives that are so clearly true negatives that an RCT is unnecessary" and "in such cases as it is doubtful an IRB would approve of such a contraindicated treatment"?
> > > > > Examples include (pulled from the data linked in our paper):
> > > > > Warfarin for hemophilia
> > > > >
> > > > > thiabendazole for liver failure
> > > > >
> > > > > thalidomide for pregnancy
> > > > >
> > > > > ceftriaxone for liver disease
> > > > >
> > > > > chloramphenicol for aplastic anemia
> > > > >
> > > > > bethanechol for IBD
> > > > >
> > > > > methoxsalen for melanoma
> > > > >
> > > > > trandolapril for hypotension
> > > > >
> > > > > diethylpropion for substance abuse
> > > > >
> > > > > dihydrocodeine for constipation
> > > > >
> > > > > etc.
> > > > >
> > > > >
> > > > > 2.	Please explain in detail how MyChem and NDF-RT generate "contraindicated for" labels for so many drug-disease pairs.
> > > > > We refer you to the peer-reviewed articles:
> > > > >
> > > > > MyChem (an API that pulls together previously peer-reviewed drug knowledge):
> > > > >
> > > > > 10.1186/s12859-018-2041-5
> > > > >
> > > > > 10.1186/s13059-016-0953-9
> > > > >
> > > > > NDF-RT (a database that indexes FDA approved/contraindicated drugs):
> > > > >
> > > > > 10.1016/j.jbi.2017.07.013
> > > > >
> > > > > 10.3233/978-1-60750-949-3-477
> > > > >
> > > > > As mentioned in the first paragraph of this response, we do not in any way intend for our proposed approach to be applied clinically or in any way in the treatment of patients. As such, we rely on aggregated, previously published results, relying on the scientific process to allow us to at least by proxy, assume that these results are somewhat sound in our effort to detail promising AI/ML directions for future methodological development.

---

> > > > > > ### Comment · Reviewer_cTJB · 2022-11-22
> > > > > > **RepoDB is a reliable source for TP/TN and can be used for evaluation**
> > > > > >
> > > > > > RepoDB is a reliable source for TP/TN (all from RCT results). That's why I suggested additional experiments in my original reviews.
> > > > > > "a. The paper uses only cross-validation (8:1:1) for a link prediction problem, which leaks information. Since RepoDB is the ONLY reliable TP/TN resource, I suggest the authors to use the other 3 datasets to predict links in RepoDB - it avoids dataset information leak."

---

> > > > > > > ### Author Response · Authors · 2022-11-28
> > > > > > > **Agree RepoDB is *a* reliable source, and will be fruitful to try as you suggest, but is not the *only* reliable source**
> > > > > > >
> > > > > > > We will certainly consider training only on RepoDB (even though it has a paucity of TN examples), though respectfully disagree that this is the “ONLY reliable TP/TN resource.” Evidence would need to be presented to support such an assertion, beyond affinity towards one repository’s design over another.

---

> ### Author Response · Authors · 2022-11-11
> **Response to Reviewer cTJB (part 3)**
>
> # weakness 4
>
> Thanks for your suggestions.
>
> For question a, if I understand your concern correctly, you consider that the data leakage comes from the duplicate drug-disease pairs from four data sources, and you also think that other 3 data sources are not reliable except for RepoDB. Actually, as we mentioned in Appx. A.3.2, we did significant  preprocessing of the raw data of 4 datasets. One of these preprocessing steps is mapping the identifier of drugs and diseases from these four data sources to the identifiers used in knowledge graph (note that the same drugs or diseases might use different identifiers from different data sources) and then remove duplicate drug-disease pairs before we split them into training set, validation set, and test set (8:1:1). This entity mapping avoids data leakage. Moreover, as we reply as above, MyChem, SemMedDB, NDF-RT are also human-curated or literature-derived datasets which contain reliable data.
>
> For question b, we mentioned “identify new indications... of drugs/compounds” because this is the definition of drug repurposing. However, we clearly define our goal in sec. 4.1 that “We solve drug repurposing as a link prediction problem on the knowledge graph … given any drug-disease pair (v_i, v_j), we predict the probability that drug i can be used to treat disease j”. Also, due to the page limit and there are many drugs, reporting the performance of predicting interactions for each drug is impossible. Instead, we reported the accuracy, F1 score, and ranking metrics of link predictions as an overall evaluation of the purpose framework.
>
> For question c, the “leave-drug-class-out” idea is a good idea for follow-up experiments. However, our model does not leverage  “drug class” information to boost its performance. Since a drug class is a set of drugs/compounds with similar chemical structure resulting in similar function to the diseases, and our model and data do not use chemical structure similarity whatsoever. Our model only utilizes the topology structure of the knowledge graph rather than drug chemical structures and the knowledge graph also doesn’t contain any “drug class” information. Therefore, the model considers the similar drugs only when they have similar neighbor structure. This leads  us to believe there should be little impact to model performance with regard to drug class, and no associated information leakage.
>
> For question d, the “probabilities” in the tables 8 and 9 mean the machine learning model considers the likelihood that a given drug might have potential to treat a given disease based on the input features extracted from the knowledge graph. Therefore, these “probabilities” can only give a “suggestion” to what drugs/compounds the pharmacologists can prioritize in a drug experiment given the training data.
>
>
> # weakness 5
> Sorry for this minor error. The references have been sorted in the updated version.

---

### Decision · Program_Chairs · 2023-01-20

**Decision:**

Reject

**Justification For Why Not Higher Score:**

Inadequate comparison with baseline methods, validation in experiments, and the limited technical novelty.

**Justification For Why Not Lower Score:**

N/A

**Metareview: Summary, Strengths And Weaknesses:**

The authors proposed an approach for drug repurposing based on a biomedical knowledge graph. It not only predicts drug efficacy against a disease, but it also finds paths that could provide a biological explanation for the predictions. They compared their method against competitors and showed their performance in predicting drugs and in recapitulating curated drug mechanism. The approach combines existing methods: PubMedBERT for defining the node attributes in the knowledge graph, GraphSAGE for learning node embeddings, random forest for predicting drug efficacy, and ADversarial Actor-Critic (ADAC) Reinforcement learning model for predicting drug mechanism of action paths.

Strength of the paper:
1. The paper is written clear with introduction to drug repurposing.
2. Experiments show that the number of false positive predictions is reduced.

Weakness of the paper:
1. The paper lacks comparison with important drug repurposing baselines.
2. Evaluation setup is questionable. The authors fix some of them (validation/testing split) in response.
3. The technical novelty is limited while the main novelty is the combination of building blocks.

In summary, the paper is not yet ready to appear in ICLR.